# Protein Structure Tokenization: Benchmarking and New Recipe

Xinyu Yuan [* 1 2]  Zichen Wang [† 3]  Marcus Collins [3]  Huzefa Rangwala [† ‡ 3]

## Abstract

Recent years have witnessed a surge in the development of protein structural tokenization methods, which chunk protein 3D structures into discrete or continuous representations. Structure tokenization enables the direct application of powerful techniques like language modeling for protein structures, and large multimodal models to integrate structures with protein sequences and functional texts. Despite the progress, the capabilities and limitations of these methods remain poorly understood due to the lack of a unified evaluation framework. We first introduce **Struct-TokenBench**, a framework that comprehensively evaluates the quality and efficiency of structure tokenizers, focusing on fine-grained local substructures rather than global structures, as typical in existing benchmarks. Our evaluations reveal that no single model dominates all benchmarking perspectives. Observations of codebook underutilization led us to develop **AminoAseed**, a simple yet effective strategy that enhances codebook gradient updates and optimally balances codebook size and dimension for improved tokenizer utilization and quality. Compared to the leading VQ-VAE model ESM3, our method achieves an average of 6.31% performance improvement across 24 supervised tasks, with sensitivity and utilization rates increased by 12.83% and 124.03%, respectively. Source code and model weights are available at https://github.com/KatarinaYuan/StructTokenBench.

---

[*]This work was completed while the author was an intern at Amazon. [†]Corresponding author. [‡]Huzefa Rangwala is on LOA as a Professor of Computer Science at George Mason University. This paper describes work performed at Amazon. [1]Mila - Quebec AI Institute [2]University of Montreal [3]Amazon. Emails: Xinyu Yuan <xinyu.yuan@mila.quebec>, Zichen Wang <wangzc921@gmail.com>, Marcus Collins <collmr@amazon.com>, Huzefa Rangwala: <rhuzefa@amazon.com>

*Proceedings of the $42^{nd}$ International Conference on Machine Learning*, Vancouver, Canada. PMLR 267, 2025. Copyright 2025 by the author(s).

## 1. Introduction

Proteins, linear sequences of amino acid residues that fold into complex 3D macromolecules, drive the essential machinery of life. While protein language models (PLMs) trained on residue sequences have emerged as powerful tools for deciphering the underlying "language" of proteins and decoding evolutionary patterns in protein function (Lin et al., 2023b), their reliance on sequence tokenization (*e.g.*, per-residue or sub-word methods like BPE (Ferruz et al., 2022)) overlooks a critical dimension: the 3D structural context. Residues adopt diverse geometric conformations to perform biological functions, and their structural interactions-not just sequence-dictate protein function and behavior (Abramson et al., 2024). To address this gap, recent work has explored protein structure tokenization (PST), which encodes local 3D context into discrete or continuous representations. Discrete structural tokens enable powerful techniques like language modeling of protein structures, and large multimodal models (Yin et al., 2023) that incorporate other related data modalities, such as protein sequences and function descriptions (Hayes et al., 2025).

Current PST methods fall into two categories: (1) **heuristic methods** relying on domain-specific knowledge and hard-coded rules to extract structural-based tokens (de Brevern, 2005; Durairaj et al., 2020); and (2) **deep learning methods** that use neural networks to learn features from protein structure data. The latter further includes: **VQ-VAE-based** (Van Den Oord et al., 2017) methods, which compress structures into discrete latent spaces in codebooks via reconstruction objective functions (Van Kempen et al., 2024; Lin et al., 2023a), and **Inverse-Folding-based** (IF-based) methods, which compress structures by predicting sequences capable of folding into the given target protein structure (Dauparas et al., 2022). Despite the emerging progress, the capabilities and limitations of these PST methods remain poorly understood due to the absence of a unified evaluation framework.

To address the gap, we present StructTokenBench, a comprehensive evaluation framework (Fig. 1) that assesses PSTs across four perspectives. Unlike existing protein structure benchmarks, which primarily focus on global structures (Townshend et al., 2021), StructTokenBench is designed to evaluate the quality of the latent space defined by the PST encoder/codebook, with an emphasis on fine-grained local protein structure representations.

By evaluating leading open-source PSTs, we observe that no single PST method dominates across all four perspectives: IF-based PSTs stand out in **Downstream Effectiveness** (Sec. 5.2), ProTokens performs best in **Sensitivity** (Sec. 5.3) and **Distinctiveness** (Sec. 5.4), while FoldSeek achieves superior **Codebook Utilization Efficiency** (Sec. 5.5). To gain deeper insights into PSTs, we conduct a series of ablation (Sec. 5.6) and scaling (Sec. 5.7) studies, which reveal the following observations:

- **Vector quantization** affects model expressiveness mainly due to optimization challenges, not the representation format (discrete or continuous).

- **Structural tokens** retain most of the information present in **amino acid tokens** but are less robust to noise.

- **Reconstruction quality** does not consistently correlate with **codebook quality**, indicating that both are essential to quantify PST quality.

- **Scaling up VQ-VAE-based PST encoders** yields sub-exponential benefits that eventually diminish, showing that scaling alone is insufficient for large improvements.

We observe that current PST methods exhibit low codebook utilization (Sec. 5.5), with codebooks being severely under-utilized (e.g., up to 70% of 4096 codes in ESM3 remain inactive during inference). This inefficiency aligns with "codebook collapse" (Zhang et al., 2024a), a well-known issue in VQ-VAEs where encoders disproportionately assign inputs to a small subset of tokens, limiting representational capacity.

To mitigate this, we further propose AminoAseed, a VQ-VAE-based PST (Sec. 4.2) that introduces two techniques: (1) **codebook reparameterization**: a targeted recipe for codebook collapse to improve codebook gradient update during optimization, and (2) **pareto-optimal codebook configuration**: a data-driven strategy to balance codebook size and dimension, maximizing token diversity while minimizing redundancy. We demonstrate that AminoAseed outperforms current VQ-VAE-based PSTs by a large margin.

## 2. Preliminaries

### 2.1. Problem Definition

With StructTokenBench evaluating PSTs' quality, we study the problem of developing effective PST methods. A protein structure is represented by its backbone 3D coordinates $x \in \mathbb{R}^{L \times N_{atoms} \times 3}$ and residue sequence $r \in \mathcal{S}^L$, where $L$ is the protein length, $N_{atoms} = 4$ refers to the backbone atoms: $[N, C_\alpha, C, O]$, and $\mathcal{S}$ denotes the set of 20 amino acid types[1].

---

[1]Our evaluation focuses on PSTs using backbone structure as input. While methods like Cheap (Lu et al., 2024) (incorporating all-atom structures and sequence data) lie outside this focus, they remain applicable for evaluation on all tasks (see App. F.6).

### 2.2. VQ-VAE-based PST Method

As illustrated in Fig. 1(a), VQ-VAE can be summarized as:

$$x \xrightarrow{\text{Structure Encoder}} z \xrightarrow{\text{Quantization}} q_k \xrightarrow{\text{Structure Decoder}} \tilde{x},$$

where (1) a structure encoder maps structure $x$ into a continuous representation $z \in \mathbb{R}^{L \times D}$; (2) a vector quantization layer discretizes each $z_i (1 \le i \le L)$ into a codebook vector $q_{k_i} \in \mathbb{R}^D$ by selecting its nearest neighbor from a learnable codebook $Q \in \mathbb{R}^{K \times D}$ using distance measure $d(\cdot)$: $k_i = \arg\min_j d(z_i, q_j)$; and (3) a structure decoder reconstructs 3D coordinates $\tilde{x}$ from the discrete codes $q_k = \{q_{k_j}\}_{j=1}^L$. To handle the non-differentiability of quantization, straight-through gradient estimation (Bengio et al., 2013) is used (see App. E.3).

**VQ-VAE objective.** VQ-VAE maximizes the log-likelihood of the data using the ELBO (Kingma, 2013):

$$\mathcal{L}_{\text{ELBO}} = \mathbb{E}_{p(q_k|x)}[\log p(\tilde{x}|q_k) - \text{KL}[p(q_k|x)||p(q_k)]], \text{ (1)}$$

where $p(q_k|x)$ comprises the deterministic encoding process $p(z|x)$ and the nearest neighbor selection, and $p(q_k)$ can be defined as a simple uniform prior. Thus, we obtain the KL divergence term as a constant, excluded from Eqn. 1.

Besides the "reconstruction loss" $\log p(\tilde{x}|q_k)$ from Eqn. 1, VQ-VAE introduces a "quantization loss" $||sg(z) - q_k||_2^2$ to learn codebook vectors , and a "commitment loss" $\beta||z - sg(q_k)||_2^2$ to pull the encoder's output towards the codebook vectors. $sg(\cdot)$ stands for stop-gradient operator and $\beta$ is a hyper-parameter, typically robust and set to a value in $[0.25, 2]$. Overall, the optimization objective becomes:

$$\mathcal{L} = \log p(\tilde{x}|q_k) + ||sg(z) - q_k||_2^2 + \beta||z - sg(q_k)||_2^2. \text{ (2)}$$

### 2.3. IF-based PST Method

As illustrated in Fig. 1(b), IF-based PST can be summarized as:

$$x \xrightarrow{\text{Structure Encoder}} z \xrightarrow{\text{Sequence Decoder}} \tilde{r},$$

where (1) a structure encoder similarly maps input structure $x$ into a continuous representation $z \in \mathbb{R}^{L \times D}$; (2) $z$ is decoded into protein's corresponding residue sequence $\tilde{r}$.

**IF objective.** Per-residue cross entropy is calculated by comparing $\tilde{r}$ with the ground truth sequence $r$.

### 2.4. Tokenization Process

In Fig. 1(c), the structure tokenization process is illustrated as follows: a pre-trained, fixed structure encoder maps the structure into discrete or continuous structural representations. Specifically, VQ-VAE-based PST encoders generate codebook indices as "structural tokens" and their corresponding latent vectors as "discrete structural representations", and IF-based PST encoders' output is directly viewed as "continuous structural representations".

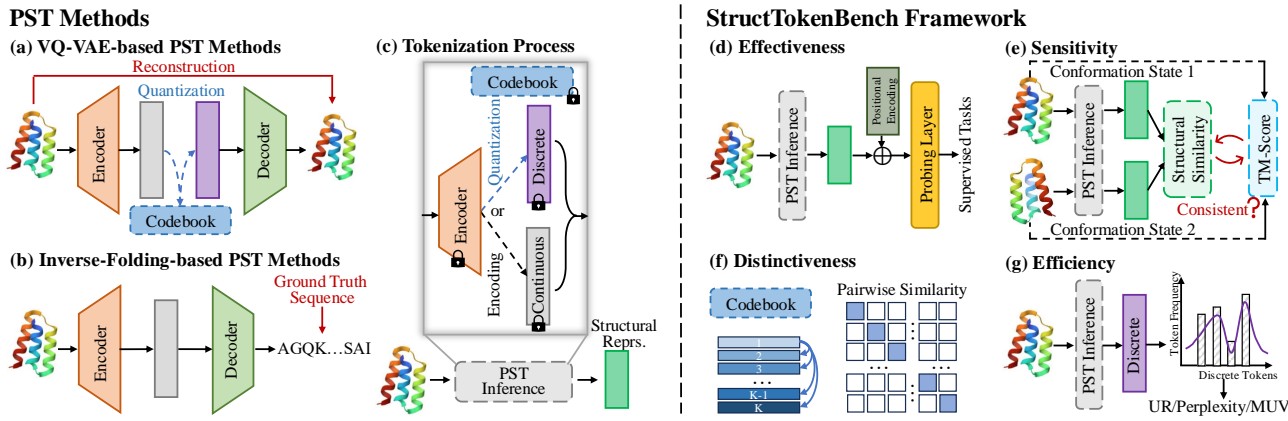

Figure 1: (a-c) Overview of PST methods, including VQ-VAE-based and IF-based methods, along with their structure tokenization process. (d-g) Overview of StructTokenBench framework, evaluating four perspectives: Downstream Effectiveness, Sensitivity, Distinctiveness, and Codebook Utilization Efficiency.

Table 1: Overview of the four perspectives in StructTokenBench, which summarizes the task types, metrics, task names and data origin.

| Perspective | Task Type | Metric | Task Name (Data Origin) |
|---|---|---|---|
| | **Functional Site Prediction** (Per-residue Binary Classification) | | |
| Downstream Effectiveness | Binding Site Prediction
Catalytic Site Prediction
Conserved Site Prediction
Repeat Motif Prediction
Epitope Region Prediction | AUROC | **BindInt** (InterPro), **BindBio** (BioLIP2), **BindShake** (ProteinShake)
**CatInt** (InterPro), **CatBio** (BioLIP2)
**Con** (InterPro)
**Rep** (InterPro)
**Ept** (PtoteinGLUE) |
| | **Physicochemical Property Prediction** (Per-residue Regression) | | |
| | Structural Flexibility Prediction | Spearman's $\rho$ | **FlexRMSF** (ATLAS), **FlexBFactor** (ATLAS), **FlexNEQ** (ATLAS) |
| | **Structure Property Prediction** (Per-protein Multiclass classification) | | |
| | Remote Homology Detection | Macro F1 | **Homo** (TAPE) |
| Sensitivity
Distinctiveness
Codebook Utilization Efficiency | Structural Similarity Consistency
Code Pairwise Similarity
Code Usage Frequency | PCC, Spearman's $\rho$
Cosine
UR, Perplexity, MUV | **Fold Switching, Apo Holo**
**CASP14**
**CASP14, CAMEO** |

# 3. Benchmark

StructTokenBench evaluates PSTs from four axes: (1) **Downstream Effectiveness** in capturing meaningful structural representations via supervised tasks (Sec. 3.1); (2) **Sensitivity** to discriminate highly similar structures (Sec. 3.2); (3) **Distinctiveness** of codebook vectors to minimize redundancy (Sec. 3.3); (4) **Codebook Utilization Efficiency** (Sec. 3.4). Datasets with corresponding task names, and evaluation metrics are described in Sec. 3.5 and Sec. 3.6, respectively.

## 3.1. Downstream Effectiveness

A proficient PST method must effectively capture critical biological details from protein structures. We evaluate this capability through supervised property prediction tasks spanning functional, physicochemical, and structural characteristics at residue and protein levels. As shown in Fig. 1(d), we integrate PST-extracted structural representations with positional encodings, and feed them into a probing layer for supervised learning. As detailed in Tab. 1, we adopt 12 tasks (24 test splits), categorized into the following 7 types.

**Task Type 1: Binding site prediction** identifies specific

amino acid residues in proteins that interact with ligands, known as binding sites (Dhakal et al., 2022). This is crucial for understanding protein functions and interactions within the cell, and it helps the development of targeted drug design and therapy (Min & Lee, 2022).

**Task Type 2: Catalytic site prediction** identifies residues that catalyze biochemical reactions in enzymes, which enhances our understanding of metabolic pathways and enzyme function. It facilitates the design of enzyme inhibitors and activators that are instrumental in drug development and therapeutic treatment (Athar et al., 2021).

**Task Type 3: Conserved site prediction** identifies residues that are evolutionarily conserved across species, which are essential to understanding the fundamental functions and structural stability of proteins. This helps to identify regions vital for protein activity, which are often targets for therapeutic drugs and genetic engineering (Boike et al., 2022).

**Task Type 4: Repeat motif prediction** detects repeated units of sequence or structural motifs within proteins, which can enhance structural stability, contribute to functional diversity, and play key regulatory roles. These motifs assist in developing biomimetic materials (Demirel, 2021).

**Task Type 5: Epitope prediction** predicts regions in proteins recognized by antibodies, known as epitopes. They are essential to understand how pathogens interact with the host immune system and to develop effective vaccines and immunotherapies (Kessler & Melief, 2007).

**Task Type 6: Structural flexibility prediction** predicts residue-level protein flexibility using metric RMSF, B-factor, and Neq (see App. B.1). Flexibility is integral for capturing protein dynamics and often relates to key functional sites, such as enzyme active sites or ligand binding sites, under varying physiological conditions (Sun et al., 2019).

**Task Type 7: Remote homology detection** detects distantly related proteins through sequence or structural similarities, which often share evolutionary origin and similar functions. This is crucial for inferring the functions of unknown proteins and tracing their evolutionary connections.

## 3.2. Sensitivity

Proteins are inherently flexible, adopting distinct conformations influenced by their environment. While these conformations may appear globally similar, they often exhibit key local structural variations that enable diverse biological functions (Chakravarty & Porter, 2022). Thus, PST methods must be sensitive to detect subtle conformational changes.

To benchmark PST sensitivity, we examine the correlation between PST-extracted structural representation similarity and the topological similarity between conformations, measured using TM-score (Zhang & Skolnick, 2005). As shown in Fig. 1(e), PSTs encode each conformation into structural representations. Given that conformations may vary in length, the representation similarity is computed via dynamic programming alignment (details in App. D.1). The correlation reflects the sensitivity of PSTs in capturing subtle differences in related protein structures.

## 3.3. Distinctiveness

VQ-VAE-based PST methods require codebook diversity to maximize the tokenizer's expressive power and reduce redundancy. Ensuring no highly similar codebook vectors also prevents ambiguous token-substructure mappings (Hayes et al., 2025), thereby eliminating confusion in downstream tasks.

We analyze the distribution of similarity between codebook vector pairs to assess the diversity of the learned codebook. Considering that codebook vectors are not uniformly utilized in practice, we also evaluate the similarity distribution weighted by token usage frequency, derived from tokenizing unseen protein structures. This offers more precise insights into the tokenizer's operational dynamics among the utilized codes in a codebook.

## 3.4. Codebook Utilization Efficiency

For VQ-VAE-based PST methods, codebook utilization reflects computational efficiency. Underutilized codebooks waste resources and harm performance, while high utilization bring gains, like LLaMa3's 128K-token vocabulary for efficient encoding (Grattafiori et al., 2024). For PSTs, for example, in language models using structural tokens, masked token prediction involves multi-class classification over the codebook and poor token utilization may yield irrelevant token predictions, degrading classification accuracy. To evaluate this, we tokenize unseen protein structures to measure utilization.

## 3.5. Datasets

We collected datasets from various resources: ATLAS (Van der Meersche et al., 2024), InterPro (Blum et al., 2024), BioLIP2 (Zhang et al., 2024b), ProteinShake (Kucera et al., 2024), ProteinGLUE (Capel et al., 2022), TAPE (Rao et al., 2019), Fold Switching (Chakravarty & Porter, 2022), Apo Holo (Saldaño et al., 2022), CAMEO (Robin et al., 2021), and CASP14 (Kryshtafovych et al., 2021) (see App. A.1.1).

As shown in Tab. 1, each dataset is linked with a task named by combining (and abbreviating) the task type, data source, and predicted property. For instance, BindInt and BindBio refer to task type "binding site prediction" using data from InterPro and BioLIP2, respectively; FlexRMSF denotes "structural flexibility prediction" and predicted property RMSF; CAMEO refers to the data source. The sizes of datasets range from tens to tens of thousands (see Tab. 6).

For supervised tasks evaluating downstream effectiveness, datasets are split using a remote homologous method (see App. A.2) to assess out-of-distribution generalization, which results in two test splits: fold (**Fold**) and superfamily (**SupFam**). Exceptions include BindShake, which retains its original test split (**Org**), and Homo, which uses its self-curated family (**Fam**), superfamily (SupFam) and fold (Fold) splits. Detailed statistics for these datasets-including length and label distributions, protein CATH structure class distribution, and sequence similarity between splits-are provided in App. A.3.

Sensitivity evaluation utilizes two datasets targeting distinct conformational behaviors: "Fold Switching" focuses on proteins that adopt multiple stable structures to enable different functions, while "Apo Holo" examines proteins undergoing conformational changes upon ligand binding to activate their biological activity.

For distinctiveness and codebook utilization efficiency, we measure structure token usage frequency on unseen proteins from CASP14 and CAMEO, which are standard holdout test sets for structure-related models, excluded from major databases.

### 3.6. Metrics

Tab. 1 outlines all metrics used in StructTokenBench: **(1) Downstream Effectiveness** is measured using AUROC for binary classification, Spearman's Rank Correlation Coefficient (Spearman's $\rho$) for regression, and Macro F1 for multiclass classification; **(2) Sensitivity** is evaluated with Pearson Correlation Coefficient (PCC) and Spearman's $\rho$ to quantify the correlation; **(3) Distinctiveness** employs cosine similarity to assess codebook diversity; and **(4) Codebook Utilization Efficiency** uses Utilization Rate (UR) and Perplexity to assess token usage, and Marginal Utility of Vocabularization (MUV) (Xu et al., 2020) to assess the trade-off of codebook entropy gain against codebook sizes.

## 4. Method

This section first outlines the motivation for our proposed method AminoAseed (Sec. 4.1), details its design innovations (Sec. 4.2), and discusses the overall pipeline (Sec. 4.3).

### 4.1. Motivation

"Codebook collapse" is a well-known issue in the VQ-VAE literature (Zhang et al., 2024a): during training, only a limited subset of code vectors are actively used, rendering others redundant. This underutilization impairs VQ-VAE's capacity to encode diverse information through the quantization bottleneck, critically hindering its model efficacy. Our benchmark, StructTokenBench, reveals analogous codebook utilization efficiency challenges in leading PSTs such as ESM3, where similar underutilization patterns emerge (see Tab. 4).

The root cause of codebook collapse stems from the challenges of training a codebook from scratch. As explained in Fig. 2, over the course of training, the distribution of learned continuous representations inevitably shifts from the entire codebook vector distribution, because a significant portion of unused codes do not receive gradient updates.

### 4.2. Method Design

To address codebook collapse, we introduce AminoAseed, a simple yet effective PST strategy employing two key techniques: (1) **Codebook Reparameterization** to enable gradient update of the entire codebook to alleviate its distribution shift, and (2) **Pareto-Optimal Codebook Configuration** to maximize token utilization while minimizing redundancy by optimally balancing codebook size and dimensions.

First, we reparameterize the codebook via a learnable linear transformation applied to fixed orthogonal vector basis: $Q = \text{Linear}(C)$, where $C$ is randomly initialized as roughly orthogonal and remains fixed during training. Unlike vanilla VQ-VAE that updates only selected codes, this allows the entire linear layer to learn and receive updates. In Sec. 5, we show that this approach outperforms vanilla VQ-VAE across

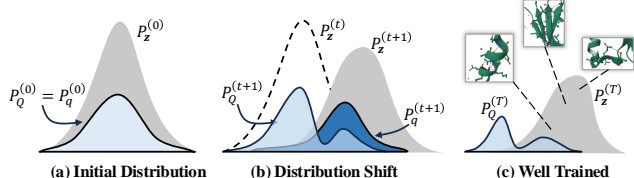

**(a) Initial Distribution**  **(b) Distribution Shift**  **(c) Well Trained**

Figure 2: Evolution of representations distributions in VQ-VAE-based PST method during training. (a) **Initial distribution**: distributions for the encoder output $P_{\boldsymbol{z}}^{(0)}$, the entire codebook vectors $P_{\boldsymbol{Q}}^{(0)}$, and the selective codes receiving gradient updates during quantization $P_{\boldsymbol{q}}^{(0)}$ are initialized. (b) **Distribution shift**: from training step $t$ to $t+1$, the distribution $P_{\boldsymbol{z}}^{(t+1)}$ updates, while unused code vectors do not update, leading to a distribution bifurcation and misalignment for $P_{\boldsymbol{Q}}^{(t+1)}$ and $P_{\boldsymbol{q}}^{(t+1)}$. (c) **Well Trained**: By the final step $T$, $P_{\boldsymbol{z}}^{(T)}$ is well trained to reflect the local 3D structures for residues.

all benchmark perspectives for various codebook sizes.

Second, we balance codebook size ($K$) and dimension ($D$) through data-driven Pareto-optimal scaling under the codebook capacity constraint ($K \times D$). Our scaling experiments (Sec. 5.7) reveal two critical trends: **(1) for downstream effectiveness**, supervised task performance degrades with extreme codebook sizes-either overly large ($K > 2^{10}$) or small ($K < 2^8$); and **(2) for codebook utilization efficiency**, it monotonically drops as $K$ increases. These findings suggest a moderate size ($K = 2^9$) that optimizes codebook utilization efficiency without sacrificing downstream effectiveness. This choice aligns with biological insights from heuristic PST method TERMs (Mackenzie, 2016), which suggests that around 600 substructures are adequate to describe 50% of the PDB database at sub-angstrom resolution.

### 4.3. Overall Pipeline

AminoAseed builds upon ESM3's local frame paradigm to model protein structures, utilizing per-residue frames and their neighboring frames to capture backbone geometry via relative distances and orientations. During encoding, we employ geometric self-attention layers to maintain rotation and translation invariance (see App. E.2). Post-quantization, discrete latent representations are decoded using standard bidirectional transformer blocks to reconstruct the structure. The training objective integrates the commitment and quantization losses in Eqn. 2, and five distinct reconstruction loss terms from ESM3. Details are discussed in App. E.1.

## 5. Experiments

Our benchmarking results are detailed as follows: downstream effectiveness in Sec. 5.2, sensitivity in Sec. 5.3, distinctiveness in Sec. 5.4, and codebook utilization efficiency in Sec. 5.5. We further conducted ablation studies for structural tokens (Sec. 5.6) and scaling studies for model config-

urations (Sec. 5.7). We also show visualizations of protein reconstruction quality and codebook vectors in App. F.5 as a case study.

## 5.1. Setups

**Pre-training dataset.** For the protein 3D structures used for pre-training, we followed the same criteria from training OpenFold2 (Ahdritz et al., 2024) to filter structures from RCSB Protein Data Bank (PDB) (wwp, 2019). We downsampled 10% of the filtered data, resulting in 48,316 protein chains with less than 40% sequence identity to each other, to train our PSTs. We split the data at ratio of 90% and 10% for training and validation sets. For held-out test sets, we use CAMEO and CASP14, which include 189 and 35 protein structures, respectively. Details are provided in App. F.1.

**Pre-training configurations.** Adam optimizer (Kingma, 2014)(learning rate: 0.0001, weight decay: 0.01, warmup steps: 5, 426) was used to train AminoAseed for 108,530 steps on 8 NVIDIA A100 GPUs (more details in App. F.1).

**Structure tokenization baselines.** We benchmarked AminoAseed against leading open-source PST methods: (1) VQ-VAE-based PSTs, including FoldSeek (Van Kempen et al., 2024), ProTokens (Lin et al., 2023a), and ESM3 (Hayes et al., 2025); and (2) IF-based PSTs, including ProteinMPNN (Dauparas et al., 2022) and MIF (Yang et al., 2023). We also trained an ablated version, VanillaVQ, designed to provide a fair comparison by only differing from AminoAseed in the proposed strategy outlined in Sec. 4.2. This ablated version essentially reduces to ESM3's PST when trained under identical dataset and configuration. Results for two more PST methods are added in App. F.6 as suggested by reviewers.

**Training and evaluation configuration for supervised downstream tasks.** A two-layer MLP probing layer is used for prediction across all tasks, trained using an Adam optimizer for 10,000 steps. During training, the structural representations extracted from PSTs—either continuous or discrete—were fixed. For all models on all tasks, we opted the checkpoint with the best learning rate based on validation set performance. More details are stated in App. F.1.

**Ablation study and scaling study experiment configurations.** We provided the configuration details in App. F.1.

## 5.2. Downstream Effectiveness Results

In Tab. 2, we first discuss VQ-VAE-based PSTs. For existing models: **(1) ESM3** consistently outperforms others across all tasks; and **(2) FoldSeek** shows limited downstream effectiveness due to its small codebook size ($K$=20) and dimension ($D$=2). For our implemented models: **(1) VanillaVQ** closely matches ESM3 in functional tasks (-0.86% relative difference), though underperforms in physicochemical and structural tasks (-9.64% and -21.33%, respectively);

**(2) AminoAseed**, our proposed method, significantly improves upon ESM3 on functional (+4.74%) and structural (+27.31%) tasks, while being comparable in physicochemical tasks (-0.27%); and **(3) overall**, VanillaVQ lags ESM3 by -5.61%, while AminoAseed surpasses it with a +6.31% average relative improvement across all 24 task splits.

We next discuss IF-based PSTs: they surpass leading VQ-VAE-bsed PSTs in functional and physicochemical properties but lag in the structural ones. We attribute this to the optimization challenge for the quantization in VQ-VAE. Our ablation studies (Sec. 5.6) support this hypothesis. Notably, AminoAseed substantially narrows this performance gap.

## 5.3. Sensitivity Results

Tab. 3 shows that: **(1) AminoAseed** outperforms all models in Apo Holo and Fold Switching datasets, with an average relative gain of +12.83% over ESM3 across all metrics; **(2) ESM3** performs second best; **(3) VanillaVQ** slightly trails ESM3 by -0.82%; **(4) IF-based PSTs** underperform most VQ-VAE-based PSTs because their training objective biases them to predict identical sequence across varied conformational structures, reducing their differentiation capability.

## 5.4. Distinctiveness Results

Fig. 3 indicates that AminoAseed and VanillaVQ displays high distinctiveness in its codebook vectors, indicated by fewer cosine similarities closer to one. However, in practical applications using CASP14 data, AminoAseed maintains high distinctiveness, while VanillaVQ does not.

## 5.5. Codebook Utilization Efficiency Results

Tab. 4 reveals four key insights: **(1) FoldSeek** reaches nearly 100% utilization with an extremely small codebook, showing uniform and balanced token usage (0.75 Perplexity); **(2) ESM3** effectively uses its large codebook for over one thousand codes, likely benefiting from its diverse pre-training data like PDB, AFDB (Varadi et al., 2024) and ESMAtlas (Lin et al., 2023b); **(3) AminoAseed**, maintaining the same codebook capacity as ESM3 ($K \times D$), achieves a 124.03% relative improvement in utilization rate, despite only using a 10% subset of PDB; and **(4) Low MUV for ESM3** suggests minimal codebook entropy gain compared to FoldSeek when trading off codebook sizes, indicating that smaller, well-optimized codebooks could be favored.

## 5.6. Ablation Study

We perform ablation studies on PST-extracted structural representations to evaluate: the trade-offs between discrete and continuous representation forms, information retention from residue sequences, and their robustness to noise.

**Discrete structural representations preserve most the information of their continuous counterparts.** Driven by findings that discretization often leads to information loss (Mousavi et al., 2024), we examine how discrete and

Table 2: Benchmark results for supervised downstream tasks. We used underlining and **bold** to highlight the best performance for IF-based PSTs and VQ-VAE-based PSTs, respectively. The relative performance difference *v.s.* ESM3 for our implemented models is included.

| Task | Split | Model | | | | | | |
| --- | --- | --- | --- | --- | --- | --- | --- | --- |
| | | IF-based PST | | VQ-VAE-based PST | | | | |
| | | ProteinMPNN | MIF | FoldSeek | ProTokens | ESM3 | VanillaVQ $_{(v.s.ESM3)}$ | AminoAseed $_{(v.s.ESM3)}$ |
| **Functional Site Prediction (AUROC%)** | | | | | | | | |
| BindInt | Fold | 51.83 | 50.38 | **53.18** | 44.66 | 44.30 | 47.25$_{(+6.66\%)}$ | 47.11$_{(+6.34\%)}$ |
| | SupFam | 94.00 | 94.56 | 46.20 | 86.05 | **90.77** | 86.71$_{(-4.47\%)}$ | 90.53$_{(-0.26\%)}$ |
| BindBio | Fold | 78.42 | 85.79 | 52.37 | 58.47 | 62.84 | 62.02$_{(-1.30\%)}$ | **65.73**$_{(+4.60\%)}$ |
| | SupFam | 81.00 | 87.27 | 52.41 | 60.47 | 65.22 | 62.92$_{(-3.53\%)}$ | **68.30**$_{(+4.72\%)}$ |
| BindShake | Org | 75.52 | 79.90 | 53.40 | 59.82 | 66.10 | 67.04$_{(1.42\%)}$ | 69.61$_{(+5.31\%)}$ |
| CatInt | Fold | 61.05 | 59.62 | 53.43 | 58.16 | 61.09 | 58.89$_{(-3.60\%)}$ | **62.19**$_{(+1.80\%)}$ |
| | SupFam | 93.40 | 96.49 | 51.41 | 83.85 | 89.82 | 85.00$_{(-5.37\%)}$ | **91.91**$_{(+2.33\%)}$ |
| CatBio | Fold | 82.49 | 85.85 | 56.37 | 56.14 | 65.33 | **67.58**$_{(+3.44\%)}$ | 65.95$_{(+0.95\%)}$ |
| | SupFam | 93.19 | 96.97 | 53.78 | 64.05 | 74.65 | 70.92$_{(-5.00\%)}$ | **87.59**$_{(+17.33\%)}$ |
| Con | Fold | 57.18 | 58.43 | 49.26 | 56.23 | 55.22 | 56.98$_{(+3.19\%)}$ | **57.23**$_{(+3.64\%)}$ |
| | SupFam | 84.68 | 92.66 | 51.39 | 74.33 | 80.53 | 74.60$_{(-7.36\%)}$ | **86.60**$_{(+7.54\%)}$ |
| Rep | Fold | 77.63 | 74.53 | 47.70 | **77.25** | 74.70 | 75.99$_{(+1.73\%)}$ | 74.97$_{(+0.36\%)}$ |
| | SupFam | 80.71 | 83.11 | 52.53 | 78.90 | 82.36 | 82.09$_{(-0.33\%)}$ | **84.57**$_{(+2.68\%)}$ |
| Ept | Fold | 62.84 | 68.78 | 54.52 | 54.69 | **63.69** | 59.28$_{(-6.92\%)}$ | 62.16$_{(-2.40\%)}$ |
| | SupFam | 64.84 | 82.98 | 50.56 | 67.52 | 61.97 | 67.24$_{(+8.50\%)}$ | **72.02**$_{(16.22\%)}$ |
| **Average** AUROC% | | 75.92 | 79.82 | 51.90 | 65.37 | 69.24 | 68.30$_{(-0.86\%)}$ | **72.43**$_{(+4.74\%)}$ |
| **Physicochemical Property Prediction (Spearman's $\rho$%)** | | | | | | | | |
| FlexRMSF | Fold | 62.37 | 59.60 | 15.35 | 13.81 | 44.53 | 44.22$_{(-0.70\%)}$ | **44.63**$_{(+0.22\%)}$ |
| | SupFam | 59.24 | 56.80 | 11.99 | 7.62 | 39.68 | 39.08$_{(-1.51\%)}$ | **40.99**$_{(+3.30\%)}$ |
| FlexBFactor | Fold | 31.88 | 34.60 | 4.17 | 6.67 | **23.60** | 22.32$_{(-5.78\%)}$ | 21.30$_{(-10.09\%)}$ |
| | SupFam | 34.56 | 35.23 | 6.97 | 5.47 | **25.80** | 23.73$_{(-7.95\%)}$ | 21.76$_{(-15.59\%)}$ |
| FlexNEQ | Fold | 69.69 | 65.32 | 5.71 | 12.98 | 45.08 | 35.95$_{(-20.25\%)}$ | **49.64**$_{(+10.12\%)}$ |
| | SupFam | 68.69 | 64.82 | 2.60 | 12.50 | 45.43 | 35.61$_{(-21.62\%)}$ | **50.15**$_{(+10.41\%)}$ |
| **Average** $\rho$% | | 54.41 | 52.73 | 7.80 | 9.84 | 37.35 | 33.49$_{(-9.64\%)}$ | **38.08**$_{(-0.27\%)}$ |
| **Structure Property Prediction (Macro F1%)** | | | | | | | | |
| Homo | Fold | 25.66 | 22.56 | 11.57 | 5.84 | **30.02** | 18.17$_{(-39.47\%)}$ | 29.87$_{(-0.50\%)}$ |
| | SupFam | 30.83 | 33.86 | 4.67 | 6.17 | 24.89 | 22.10$_{(-11.21\%)}$ | **38.38**$_{(+54.20\%)}$ |
| | Fam | 63.33 | 74.22 | 15.30 | 18.33 | 54.42 | 47.18$_{(-13.30\%)}$ | **69.78**$_{(+28.22\%)}$ |
| **Average** Macro F1% | | 39.94 | 43.55 | 10.51 | 10.11 | 36.44 | 29.15$_{(-21.33\%)}$ | **46.01**$_{(+27.31\%)}$ |

continuous representations perform in supervised tasks. We present results for the challenging Fold test on half of the 24 task splits, with remaining splits reported in App. F.3.

Fig. 4 shows that continuous representations even reduce performance across many tasks, though they enhance specific tasks like FlexNEQ (Fold split). Overall, the performance differences between discrete and continuous forms remain minor, suggesting that the continuous format is not the major contributors to performance variance. This further clarifies the advantage of IF-based PSTs over VQ-VAE-based PSTs in supervised tasks (see Tab. 2). IF-based PSTs, which use a continuous representation format and lack a quantization process, likely benefit most from avoiding the complexity associated with quantization optimization.

**Structural representations encapsulate most of the information present in sequence tokens.** Tab. 5 shows that combining amino acids with structure representations helps to improve performance for most VQ-VAE-based PSTs on most task splits. Top-performing methods like AminoAseed and ESM3 witness relative small gains, indicating that their structural representations largely encapsulate the information provided by amino acids.

**Structural representations are less robust compared to sequence tokens.** Fig. 5 illustrates that PST-extracted structural representations become less reliable when exposed to noise (see App. F.1). Specifically, IF-based PSTs show reduced robustness compared to VQ-VAE-based PSTs. Moreover, using amino acid sequences alone for prediction proves more resilient than top performing PSTs. As the noise level increases to very high levels (up to 90%), the performance discrepancies among most methods diminish.

### 5.7. Scaling Study

We performed scaling studies to pretrain AminoAseed and its ablated version VanillaVQ, to understand the optimal allocation of codebook sizes and dimensions, and VQ-VAE's scaling behavior with respect to the encoder capacities.

**Reconstruction quality does not correlate with codebook quality.** We varied the codebook size $K$ and dimension $D$ while keeping the codebook capacity ($K \times D = 2^{19}$), and encoder and decoder sizes consistent with ESM3. In Fig. 6, across various codebook sizes, we observe that $K = 2^{14}$ achieves optimal reconstruction quality under a large compute budget for both models. However, codebook utilization significantly decreases for sizes larger than $2^{11}$ and the per-

Table 3: Sensitivity evaluation on conformational proteins. The relative performance difference *v.s.* ESM3 is included.

| Model | Apo Holo | | Fold Switching | |
|---|---|---|---|---|
| | PCC% | Spearman's $\rho$% | PCC% | Spearman's $\rho$% |
| ProteinMPNN | 35.80 | 45.22 | 51.46 | 55.91 |
| MIF | 35.82 | 43.55 | 54.48 | 59.27 |
| FoldSeek | 34.79 | 43.03 | 51.88 | 56.54 |
| ProTokens | **43.32** | 54.05 | 61.20 | 65.90 |
| ESM3 | 39.76 | 50.97 | 57.12 | 62.23 |
| VanillaVQ$_{(v.s.ESM3)}$ | 39.62$_{(-0.35\%)}$ | 50.61$_{(-0.71\%)}$ | 56.51$_{(-1.07\%)}$ | 61.51$_{(-1.16\%)}$ |
| AminoAseed$_{(v.s.ESM3)}$ | 42.47$_{(+6.82\%)}$ | **54.88**$_{(+7.67\%)}$ | **65.61**$_{(+14.86\%)}$ | **75.89**$_{(+21.95\%)}$ |

Table 4: Codebook utilization efficiency evaluation on CASP14 and CAMEO datasets.

| Model | #Code ($K$) | Dim ($D$) | CASP14 | | | CAMEO | | |
|---|---|---|---|---|---|---|---|---|
| | | | UR% | Perplexity | MUV | UR% | Perplexity | MUV |
| FoldSeek | 20 | 2 | **99.00** | **0.7548** | / | 100.00 | **0.7435** | / |
| ProTokens | 512 | 32 | 69.88 | 0.5369 | 2.53e-4 | 75.56 | 0.5697 | 2.51e-4 |
| ESM3 | 4096 | 128 | 27.60 | 0.2489 | 3.28e-5 | 32.10 | 0.2841 | 3.26e-5 |
| VanillaVQ | 512 | 1024 | 5.55 | 0.0339 | **2.64e-4** | 5.60 | 0.0337 | **2.62e-4** |
| AminoAseed | 512 | 1024 | 64.45 | 0.4946 | 2.54e-4 | 68.87 | 0.5119 | 2.52e-4 |

Table 5: Performance comparison of combining amino acid tokens with structural tokens versus using structural tokens alone across 24 supervised task splits. The reported metrics include: #TaskImpr (number of tasks improved more than 0.005), #TaskComp (number of comparable tasks within $\pm0.005$ difference), AvgDiff (average difference), and AvgRelDiff (average relative difference).

| Model | #TaskImpr | #TaskComp | AvgDiff% | AvgRelDiff |
|---|---|---|---|---|
| ProteinMPNN | 5 | 3 | -1.55 | -4.72% |
| MIF | 0 | 13 | -1.15 | -2.79% |
| FoldSeek | 6 | 7 | -0.42 | -13.32% |
| ProTokens | 20 | 0 | 4.43 | 29.60% |
| ESM3 | 15 | 2 | 2.39 | 4.33% |
| VanillaVQ | 18 | 2 | 3.36 | 6.98% |
| AminoAseed | 13 | 5 | 1.33 | 1.42% |

formance of supervised tasks displays a U-shape across codebook sizes, favoring size in $[2^8, 2^{10}]$ (more results in Fig. 12). This indicates that reconstruction quality does not correlate with PSTs' codebook quality, offering a different angle compared to existing PSTs like ESM3 that only use reconstruction quality as justifications for PST quality.

**VQ-VAE-based PST methods scale sub-exponentially with data and compute.** We analyzed the scaling effects of varying encoder sizes while keeping the codebook size and dimension, and decoder size constant. As illustrated in the upper right panel of Fig. 6, U-shaped IsoFLOP curves (Hoffmann et al., 2022) emerge at lower compute budgets. In Fig. 7, we observe that reconstruction loss decreases sub-exponentially when increasing training data and compute, suggesting diminishing returns with more resources. Both figures indicate that simply scaling up the encoder may not be an effective approach to improve VQ-VAE.

# 6. Related Work

## 6.1. Protein Representation Learning Benchmarks

In protein representation learning, several benchmarks evaluate protein sequence modeling: TAPE (Rao et al., 2019) with five supervised tasks, FLIP (Dallago et al., 2021) for protein fitness landscapes, PEER (Xu et al., 2022) for multi-

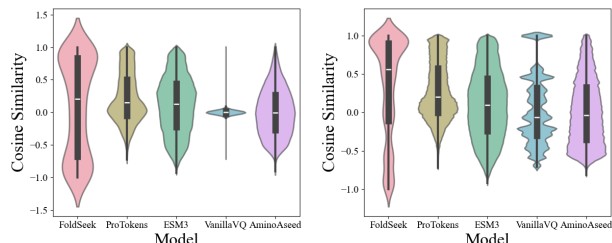

Figure 3: Distinctiveness analysis of PST codebook vectors: Left panel shows pairwise similarities between vectors; Right panel shows frequency-weighted similarities based on CASP14 token usage.

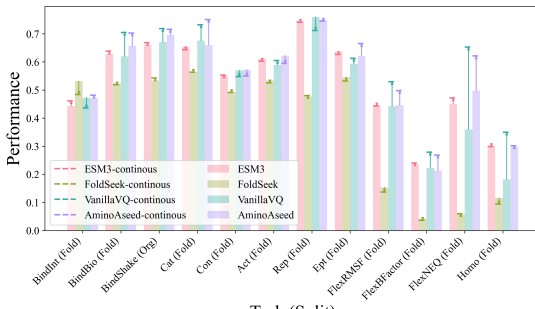

Figure 4: Performance comparison of using continuous versus discrete structural representations on supervised task splits.

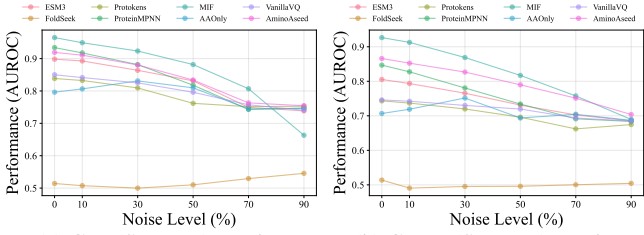

(a) Con (SupFam) results.    (b) CatInt (SupFam) results.

Figure 5: Supervised task performance with increasing noises in PST-extracted structural representations.

task benchmarking, and ProteinGLUE (Capel et al., 2022) focusing on per-residue tasks. However, benchmarks for protein structure are limited. For example, ATOM3D (Townshend et al., 2021) predicts per-protein properties for protein structures, and ProteinWorkshop (Jamasb et al., 2024) benchmarks protein structure learning tailored for geometric graph neural networks. Our work, StructTokenBench, is designed to evaluate protein structure tokenization methods, enriching the landscape of structure-related benchmarks.

## 6.2. Protein Structure Tokenization Methods

Tokenization of protein 3D structures enables efficient similarity search and cross-modality modeling of proteins with related data modalities. Early heuristic methods use domain-specific knowledge such as dihedral angles (de Brevern, 2005), secondary structure (Mackenzie, 2016), and moment invariants (Durairaj et al., 2020). Learnable PSTs include IF-based (Dauparas et al., 2022; Yang et al., 2023) and VQ-VAE-based methods (Van Kempen et al., 2024; Heinzinger

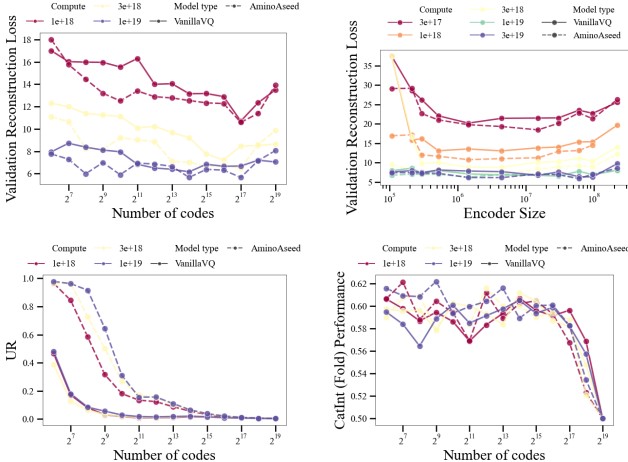

Figure 6: Impact of codebook and encoder sizes on PST quality: Top panels display reconstruction losses versus varying codebook sizes (left) and encoder sizes (right); Bottom panels shows Codebook Utilization Efficiency (left) and Downstream Effectiveness (right) performance versus codebook sizes.

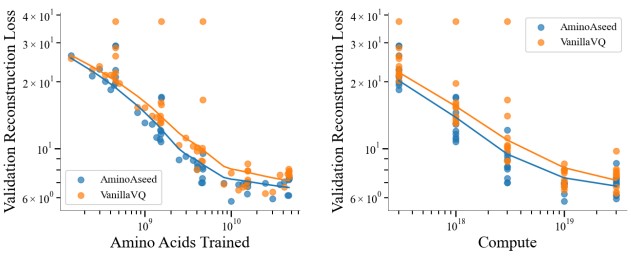

Figure 7: Scaling impact of data size (left) and compute budget (right) on reconstruction loss.

et al., 2023; Su et al., 2023; Hayes et al., 2025) (see App. G). Despite recent advances in protein structure tokenization, few studies have comprehensively compared these methods from diverse aspects. Zhang et al. (2024c) evaluated several PSTs, highlighting a trade-off between structure reconstruction and retrieval. However, this study overlooked other applications like local functional site prediction and sensitivity to structural variations, which are critical for predicting binding in variants (Loux et al., 2024).

## 7. Conclusions and Future Work

In this work, we first developed StructTokenBench, a comprehensive benchmark to evaluate protein structure tokenization (PST) methods across four key perspectives: downstream effectiveness, sensitivity, distinctiveness, and codebook utilization efficiency. StructTokenBench curated 10 public datasets about protein structure and functions, covering 17 tasks, making it the first benchmark for PST and a leading benchmark resource for protein structure representation learning. Next, we evaluated five popular state-of-the-art PST methods and found that inverse-folding-based PSTs excel in downstream effectiveness but suffer from low sensitivity, whereas VQ-VAE-based PSTs are more sensitive to protein conformations and exhibit varied efficiency

in codebook utilization. Nevertheless, no single model leads across the benchmark. Finally, we present our novel method AminoAseed with its superiority across all benchmarking perspectives using codebook reparameterization and Pareto-optimal codebook configuration.

## Acknowledgments

The authors would like to thank Zuobai Zhang, Yanjun Qi and Tommi Jaakkola for their helpful discussions and comments. We also appreciate all anonymous reviewers for their constructive suggestions.

## Impact Statement

This paper advances machine learning techniques for protein structure representation, with potential applications in computational biology, drug discovery, and protein engineering. While our research could contribute to understanding protein mechanisms and developing therapeutic interventions, these are speculative outcomes requiring further development. Our computational method does not directly involve human subjects or biological experiments. The potential societal impact lies in providing a more robust methodology for understanding protein structures, which could indirectly support medical and biotechnological research. Responsible deployment of our work and transparency in its development and use are crucial to ensure it benefits society while minimizing risks.

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

# A. Data Pre-processing

## A.1. Data Sources

### A.1.1. DOWNSTREAM EFFECTIVENESS EVALUATION

- **ATLAS** (Vander Meersche et al., 2024) database provides a comprehensive collection of standardized molecular dynamics (MD) simulations of protein structures, featured with detailed analysis of both global protein behavior and local flexibility of the protein backbone. It includes 1390 protein chains in total, each subjected to three replicated all-atom MD trajectories using GROMACS (Abraham et al., 2015) and the CHARMM36m force field (Huang et al., 2017). For our studies, we utilized the initial release of this database, identified as "2022_06_13 v1".

- **InterPro** (Blum et al., 2024) database consolidates various datasets into a single searchable platform, offering runctional insights into protein sequences by classifying them into families and identifying key domains and functional sites. We used "release 100.0" dated 30th May 2024.

- **BioLIP2** (Zhang et al., 2024b) is a semi-manually curated database that provides high-quality, biologically relevant ligand-protein binding interactions, validated through geometric rules and empirical literature. It enriches entries with detailed information, including catalytic sites and binding affinities, sourced from diverse databases and comprehensive manual literature reviews. Notably, BioLIP2 offers unique data not covered by InterPro, making it an essential alternative resource for predicting binding and catalytic sites.

- **ProteinShake** (Kucera et al., 2024) is a benchmarking software package designed to simplify dataset creation and model evaluation for deep learning applications focused on protein structures. It addresses a wide range of biological challenges, including structure-function relationships, geometric relationships between structures, and modeling physical interactions. Notably, its binding site prediction task leverages data from the PDBbind-CN database (Liu et al., 2015) and the DUDE-Z virtual screening benchmark (Stein et al., 2021), providing an alternative choice for binding site prediction besides InterPro and BioLIP2.

- **ProteinGLUE** (Capel et al., 2022) is a benchmark designed to evaluate protein representations, focusing on a variety of tasks related to structural protein properties. It includes specific tasks like protein-protein interactions and epitope mapping, which explore how proteins interact with other molecules within their environment. These tasks are critical for deepening our understanding of protein function, providing insights into the complex dynamics of molecular interactions and their implications for biological processes like immune response.

- **TAPE** (Rao et al., 2019) is a comprehensive benchmark consisting of five biologically relevant tasks designed for the semi-supervised learning of protein biology. TAPE highlights three major areas: structure prediction, detection of remote homologs, and protein engineering for protein landscapes. This benchmark serves as a crucial tool for driving progress in understanding and manipulating proteins for various scientific and medical applications. Specifically, the remote homology detection task is derived from the SCOP 1.75 database (Andreeva et al., 2020) of hierarchically classified protein domains.

  Notably, the original dataset consists of 1195 labels. We reduce the number of label classes to degrade the task difficulty, since the dataset is extremely imbalanced with limited prediction model capacity of a 2-layer MLP probing layer. Specifically, we filtered label class that has less than 50 protein samples in the training dataset, reducing from 1195 labels to 45 labels.

### A.1.2. SENSITIVITY EVALUATION

- **Fold Switching** (Chakravarty & Porter, 2022) includes 74 pairs of fold-switching proteins, which contain regions capable of adopting distinct stable secondary and tertiary structures under varying cellular conditions, or alternating between two stable folds at equilibrium. These protein pairs exhibits extremely high levels of sequence identity (mean 99%, and median 100%), yet display significantly different structures (mean TM-scores of 0.58, and median TM-scores 0.63). This provides a solid foundation for benchmarking sensitivity of PST methods to detect subtle yet critical structural variations, essential for understanding dynamic protein behaviors in varying biological contexts.

- **Apo Holo** (Saldaño et al., 2022) provides a comprehensive dataset for studying ligand-induced conformational changes in proteins. It includes 90 pairs, each consisting of an apo conformer (unbound state) and its corresponding holo form (bound to a biological relevant ligand). This dataset spans a wide spectrum of conformational diversity, quantified by the pairwise global $C_\alpha$-RMSD between their conformers, with values ranging from 0 to as much as 15. The extensive range offers a robust foundation for detailed evaluation of protein structural conformational changes.

## A.1.3. DISTINCTIVENESS & CODEBOOK UTILIZATION EFFICIENCY EVALUATION

- **CASP14 (Kryshtafovych et al., 2021)**, *i.e.*, Critical Assessment of protein Structure Prediction, serves as an independent platform for assessing various methods of protein structure modeling. provides an independent platform for evaluating protein structure modeling methods. During the assessment period, unknown protein structure sequences were posted for modeling, with submissions collected and evaluated as experimental coordinates became available. This process systematically assesses the predictive capabilities of current modeling techniques. We selected proteins released after our pre-training data cutoff date (May 2020) from CASP14 test sets.

- **CAMEO (Robin et al., 2021)**, *i.e.*, the Continuous Automated Model EvaluatiOn (CAMEO) platform, operates automated blind evaluations to complement the biennial CASP experiments. CAMEO leverages weekly prereleases of protein sequences that are scheduled for publication in the Protein Data Bank. This platform has been particularly useful in the prediction of complex protein structures, which has been observed challenging in CASP14. We used the release from April 2022 to June 2022.

### A.2. Remote Homologous Data Splitting

For the 10 datasets collected from InterPro, BioLIP2, ATLAS and ProteinGLUE for supervised downstream tasks, we followed DeepSF (Hou et al., 2018) for a strict protein splitting method to remove the remote homologous protein redundancy, by grouping proteins according to their fold and superfamily classes retrieved from CATH database (Pearl et al., 2003).

**Remote homologous relationship classification.** The classifications of remote homologous relationships comprise three levels: family, superfamily, and fold:

- **Family** groups proteins that share clear evolutionary relationships, which can be detected by common sequence comparison tools. This is **a relatively close level** of homology.

- **Superfamily** groups proteins that are more distantly related, where the homology may only be apparent through structural similarities and conserved functional sites, rather than sequence similarity. This represents **a more remote homologous** relationship than family.

- **Fold** groups proteins based on broad global structural features, often grouping multiple superfamilies. Homology at this level is **the most remote**, as the relationship may primarily be in terms of overall structural

architecture rather than sequence or functional conservation.

**Why remote homologous redundancy important?** Remote homologous redundancy focuses on structure similarity rather than sequence similarity. Intuitively, protein properties are influenced more by structures in many aspects: (1) **function**: remote homologs often retain similar biomedical functions despite low sequence similarity. This can include enzymatic activity (catalytic cite prediction in BioLIP2), ligand binding (binding site prediction in ProteinShake), or interactions with other biomolecules; (2) **stability**: proteins with similar folds may have conserved core residues that contribute to maintaining structural integrity, even if their sequences have diverged significantly; (3) **dynamics**: the dynamic behavior of protein structures, including their flexibility and conformational changes, can be conserved among remote homologs, contributing to their functional similarities.

**Method.** To eliminate homologous protein redundancy between training and test datasets, we adopted DeepSF's (Hou et al., 2018) multi-level redundancy removal approach, where we operate at two levels more related to structures: fold and superfamily levels. In this hierarchy, fold represents the broadest category, followed by superfamily. These two levels are defined as follows: at the superfamily level, proteins from the same superfamily appear in both the training and test datasets; at the fold level, no proteins from the same superfamily are shared between the sets, though proteins from the same fold may be present in both.

The splitting method involves several steps: First, we filter proteins curated from raw data without the target functional labels. Next, the fold and superfamily labels are assigned to proteins using the CATH database (Pearl et al., 2003). For each fold, superfamilies are split into two groups (60% for training and 40% for testing), creating the fold test split. For the split training data, 80% of the proteins in each superfamily are placed in training, with the remaining 20% in testing, creating superfamily-level datasets. Lastly, 20% of the test data is randomly selected to form a validation set.

### A.3. Supervised Downstream Data Statistics Analysis

**Data Sizes.** Tab. 6 shows the number of protein samples across all downstream datasets used in StructTokenBench.

**Length distribution.** Fig. 8, shows the distribution of protein lengths for the training splits of all supervised downstream datasets, visualized using histograms with kernel density estimation. Protein lengths exhibit significant variability across datasets, ranging from short sequence (less than 200 residues) to long sequence (up to 600 residues). For binding site or catalytic site predictions, different dataset sources show distinct protein length distributions, highlight-

Table 6: Data size (*i.e.*, the number of protein samples) for all datasets used in StructTokenBench.

| Dataset | Split | | | |
|---|---|---|---|---|
| | Train | Valid | Fold Test | SupFam Test |
| BindInt | 1353 | 256 | 671 | 273 |
| BindBio | 12566 | 2678 | 8112 | 2510 |
| CatInt | 3279 | 443 | 1090 | 674 |
| CatBio | 4406 | 667 | 1815 | 889 |
| Con | 7447 | 1177 | 3262 | 1497 |
| Rep | 690 | 478 | 1789 | 143 |
| Ept | 68 | 21 | 16 | 23 |
| FlexRMSF | 643 | 81 | 225 | 104 |
| FlexBFactor | 643 | 81 | 225 | 104 |
| FlexNEQ | 643 | 81 | 225 | 104 |
| | Train | Valid | Org Test | |
| BindShake | 1286 | 308 | 242 | |
| | Train | Valid | Fold Test / SupFam Test / Fam Test | |
| Homo | 6003 | 241 | 232 / 455 / 788 | |
| | | No Split | | |
| Fold Switching | | 74 | | |
| Apo Holo | | 90 | | |
| CASP14 | | 35 | | |
| CAMEO | | 189 | | |

Table 7: Protein sequence similarity across splits for downstream supervised tasks, measured with sequence identify using MMseqs2 for pairwise sequence alignment.

| Dataset | Pair of Splits (Sequence Identity%) | | | |
|---|---|---|---|---|
| | Train *v.s.* Valid | Train *v.s.* Fold Test | Train *v.s.* SupFam Test | Fold Test *v.s.* SupFam Test |
| BindInt | 45.68 | 44.08 | 45.34 | 45.19 |
| BindBio | 36.29 | 36.22 | 36.53 | 36.04 |
| CatInt | 38.66 | 39.61 | 40.08 | 47.40 |
| CatBio | 34.12 | 34.00 | 34.39 | 34.30 |
| Con | 37.02 | 38.82 | 36.94 | 37.09 |
| Rep | 37.81 | 38.37 | 38.03 | 38.44 |
| Ept | 37.34 | 36.22 | 36.53 | 36.27 |
| FlexRMSF | 38.18 | 37.83 | 37.89 | 38.08 |
| FlexBFactor | 38.18 | 37.83 | 37.89 | 38.08 |
| FlexNEQ | 38.18 | 37.83 | 37.89 | 38.08 |
| | Train *v.s.* Valid | | Train *v.s.* Org Test | |
| BindShake | 35.10 | | 35.01 | |
| Homo | 39.96 | | 40.24 | |

ing the importance of incorporating diverse data sources for the same task.

**Protein CATH structure class distribution.** CATH (Pearl et al., 2003) is a protein structure classification system that categorizes proteins domains based on their structural and evolutionary relationships. It employs a hierarchical labeling system, with the top two levels comprising, "Class", which represents the overall secondary structure content of protein domains (*e.g.*, alpha helices, beta sheets, or mixed alpha-beta structures); and "Architecture", which describes the overall arrangement and orientation of secondary structures within the protein (*e.g.*, sandwich, barrel, or roll).

Fig. 9 presents the CATH structural class distribution for the training data of all supervised downstream datasets. While most datasets show a predominance of alpha-beta structures, others, such as BindInt and Con, demonstrate a more balanced and diverse representation of "Class" across alpha, beta, and mixed structures. Notably, irregular and special structures are present in 9 out of 12 datasets. These observations emphasize the structural diversity in the StructToken-Bench atasets, providing a realistic benchmarking scenario

Table 8: Binary label distributions for all per-residue supervised downstream tasks except physicochemical property prediction (regression) and structure property prediction (per-protein multi-class classification).

| Dataset | Per Dataset | | Per Protein | |
|---|---|---|---|---|
| | Ratio of Label One% | #Total Labels | Average Ratio of Label One% | Average #Total Labels |
| BindInt | 7.44 | 253058 | 9.66 | 187.0 |
| BindBio | 2.57 | 3330120 | 3.35 | 265.0 |
| BindShake | 17.16 | 355297 | 20.11 | 276.3 |
| CatInt | 4.82 | 827918 | 5.39 | 252.5 |
| CatBio | 1.77 | 1463891 | 1.91 | 332.2 |
| Con | 8.91 | 1599141 | 12.00 | 214.7 |
| Rep | 29.84 | 95611 | 39.42 | 138.6 |
| Ept | 20.46 | 16899 | 23.51 | 248.5 |

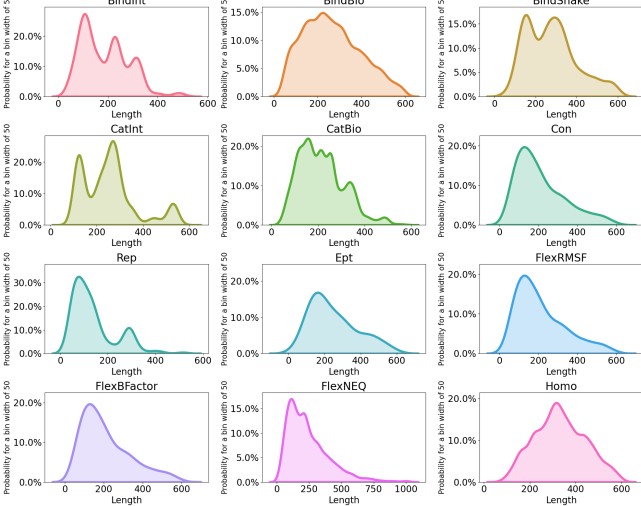

Figure 8: Length distribution for the training dataset of the 12 supervised downstream tasks.

with protein variability.

**Protein sequence similarity between splits.** The difficulty of supervised tasks largely depends on shifts in data distribution across training, validation, and different test splits (fold split and superfamily split). To effectively understand these distribution shifts, we calculate pairwise protein sequence similarities using MMseq2 (Steinegger & Söding, 2017) for sequence alignment. Sequence identity, expressed as a percentage, measures the ratio of identical amino acids to the total number of amino acids in the aligned sequences. Commonly, this metric is conventionally used to measure sequence similarity, with thresholds of 90% and 50% typically distinguishing highly similar and moderately similar sequences, respectively. According to our analysis presented in Tab. 7, the similarities among the splits range from a minimum of 34.00% to a maximum of 47.40%, with an average of 38.25%. These results suggest that the splits are sufficiently distinct to ensure that our StructTokenBench provides benchmarking tasks with reasonable difficulty.

**Label distribution for supervised binary classification tasks.** Most tasks in StructTokenBench ocus on per-residue binary classifications, where a label of one indicates functionally important residues. Typically, these functional sites

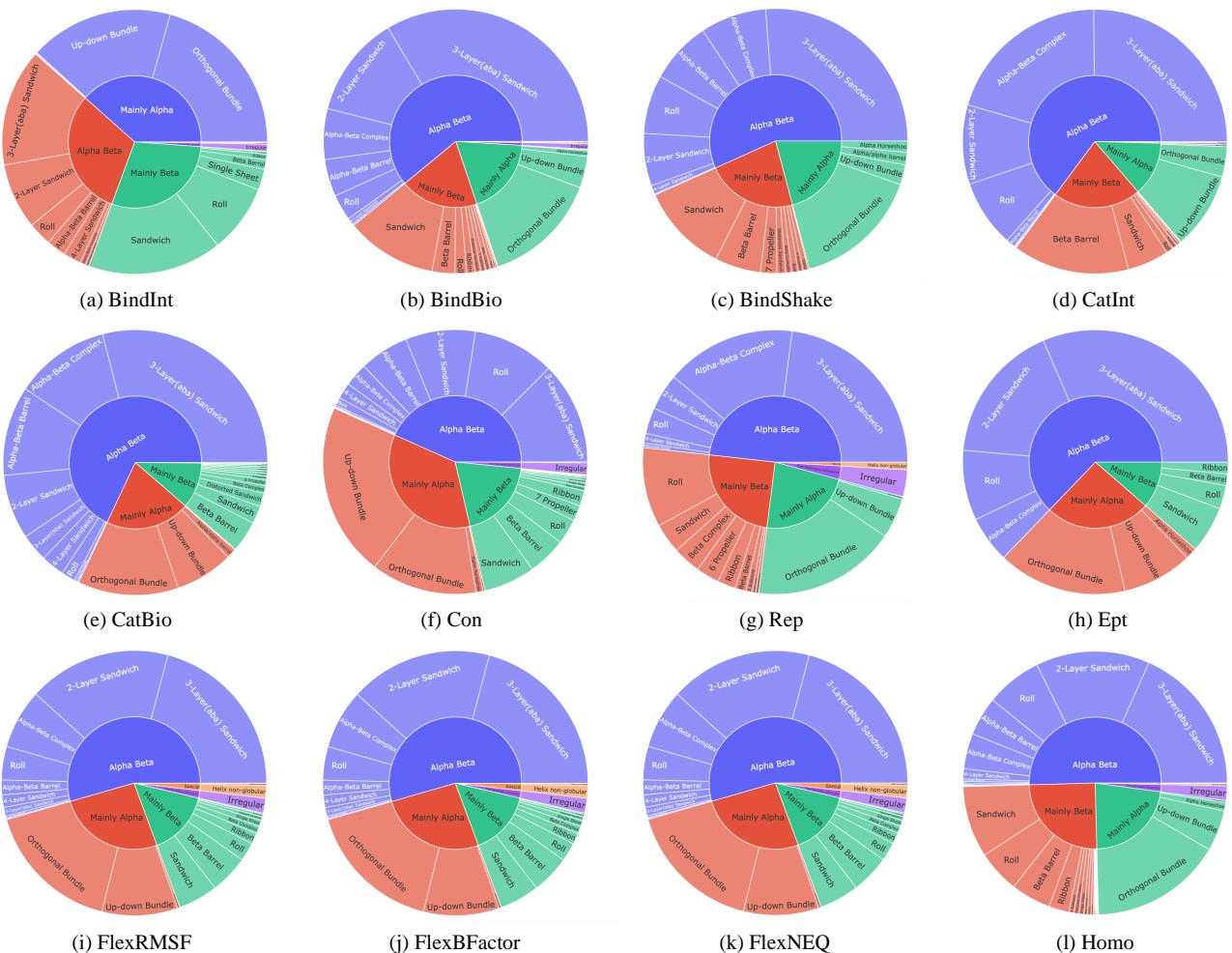

(a) BindInt     (b) BindBio     (c) BindShake     (d) CatInt

(e) CatBio     (f) Con     (g) Rep     (h) Ept

(i) FlexRMSF     (j) FlexBFactor     (k) FlexNEQ     (l) Homo

Figure 9: CATH structure class distribution for all supervised downstream datasets. Each plot represents the hierarchical breakdown of protein domains by their structural class ("Class"), and architecture ("Architecture"). Blue highlights the largest portion of class, with red the second and green the third.

are short within a single protein. To highlight this challenge, we visualized the binary label distribution, as shown in Tab. 8. This visualization confirms that labels are highly imbalanced across all tasks. To mitigate this issue, we implemented the per-batch class weighting technique in our supervised benchmarking pipeline. This method assigns different weights to the positive and negative classes, enhancing the model's capability to learn effectively from infrequent labels.

# B. Metrics

## B.1. Local Structure Flexibility Measurement in ATLAS

Certain atoms or residues in a protein tend to be more flexible and mobile, particularly in regions like enzyme active sites or protein-protein interaction interfaces, where flexibility is crucial for biological function.

- **B-Factor**, extracted from PDB files, measures how much individual atoms deviate from their average positions, reflecting both atom vibration and structural disorder. This provides insights into the structural dynamics, stability, and functional flexibility of the protein.

- **RMSF** (Root Mean Square Fluctuation), calculated on $\alpha$-carbons using GROMACS (Abraham et al., 2015), measures the average deviation of an atom from its mean position over time. It highlights regions of the protein that are either highly mobile or rigid, with higher RMSF values indicating flexible, dynamic areas, and lower values representing structural stability.

- **NEQ** quantifies the local deformability of the protein backbone by measuring the average number of protein blocks (PBs) at a given sequence position, varying from

1 (no variation) to 16 (fully random PB distribution). This is determined by the Phi/Psi angles and reflects the degree of conformational flexibility within the protein.

### B.2. StructTokenBench Metrics

Both metrics for evaluation and metrics used in the benchmarking pipeline are stated in this section.

**Downstream Effectiveness Metrics.** AUROC, Spearman's $\rho$ and Macro F1 are reported for supervised downstream performance.

- **AUROC** (Area Under the Receiver Operating Characteristic Curve) measures the ability of a classification model to distinguish between classes across all possible classifying thresholds. An AUROC of 1.0 is a perfect score, while 0.5 suggests no better than random guessing. Higher AUROC represent better performance.

- **Spearman's** $\rho$ (Spearman's Rank Correlation Coefficient) measures the rank correlation, specifically how well the relationship between two variable can be described using a monotonic function. For its computation, each observed data point is converted into ranks, with tied values receiving the average of their ranks. Spearman's $\rho$ is calculated as: $\rho = 1 - \frac{6 \sum d_i^2}{n(n^2-1)}$, where $d_i$ is the difference between ranks of corresponding variables, and $n$ is the number of observations. $\rho$ ranges from -1 to 1, where 1 indicates a perfect positive rank correlations.

- **Macro F1** calculates the average of F1 score across all classes, treating each class equally regardless of its frequency in the dataset. This makes it particularly useful in scenarios where class imbalances might distort the accuracy. The F1 score for each class is computed using the harmonic mean of precision and recall, where precision is the ratio of correctly predicted positive observations to the total predicted positives, and recall is the ratio of correctly predicted positive observations to all observations in the actual class.

**Sensitivity Metrics.** Given a pair of protein conformations, structural similarity is measured using TM-score or the negative value of RMSD (see App. F.2). Sensitivity performance is then assessed by examining the correlation between the cosine similarity of structural representations extracted by PSTs and the structural similarity of input protein conformations. This correlation is measured using PCC and Spearman's $\rho$.

- **PCC** (Pearson Correlation Coefficient) is a statistical measure that quantifies the linear relationship between two variable $X$ and $Y$, defined using the formula:

$PCC_{X,Y} = \frac{Cov(X,Y)}{\sigma_X \sigma_Y}$ where $Cov(X,Y))$ is the covariance between variables X and Y, and $\sigma_X$ and $\sigma_Y$ are the standard deviations of X and Y, respectively. PCC is widely used to assess the strength and direction of a linear relationship between two continuous variables.

- **Spearman's** $\rho$ (Spearman's Rank Correlation Coefficient) is introduced above in "Downstream Effectiveness Metrics".

- **TM-score** (Template Modeling Score), assesses the structural similarity between protein structures irrespective of their size. TM-score normalizes the score by the length of the proteins, providing a scale-invariant measure that ranges from 0 to 1, where 1 indicates a perfect match between two structures. The TM-score is less sensitive to local variations and more reflective of the overall topology of the protein structures, making it a more robust metric for comparing significantly different sizes and alignments.

- **RMSD** (Root Mean Square Deviation), measures the average distance between the atoms (usually the backbone atoms) of superimposed proteins. It quantifies the absolute spatial deviation between two aligned protein structures, providing a straightforward measure of the structural difference. Lower RMSD values indicate a higher degree of similarity between the two structures. However, RMSD is sensitive to outliers and can be heavily influenced by local discrepancies in the structure, making it less useful for comparing proteins of different sizes or those that only share partial similarity.

**Distinctiveness Metrics** Pairwise cosine similarities between codebook vectors and its weighted version based on the structural token usage frequency are reported.

- **Cosine Similarity** between two vectors $A$ and $B$ quantifies how similar the directions of the two vectors are. It is calculated as $\frac{A \cdot B}{||A|| \cdot ||B||}$, where $A \cdot B$ is the dot product, and $||A||$ and $||B||$ are the vector norms.

- **Token Usage Frequency** measures the frequency of codebook tokens being used during the protein structure quantization process.

- **Weighted Cosine Similarity** is adapted to emphasize the importance of cosine similarity for codebook vectors in practical applications. This metric is calculated in three steps: (1) calculate pairwise cosine similarity between codebook vectors; (2) determine token usage frequency in test data and calculate the pairwise product of these frequencies; and (3) weight each entry from the cosine similarity matrix by the corresponding pairwise frequency product.

**Codebook Utilization Efficiency Metrics.** UR, Perplexity and MUV are reported to understand PST codebook utilization efficiency.

- **UR (Utilization Ratio)** measures the hit ratio of codebook tokens. UR is different from token usage frequency. If one codebook token is hit during the protein structure quantization process, it's counted once towards the UR, regardless of multiple occurrences.

- **Token Usage Frequency** is introduced above in "Distinctiveness Metrics".

- **Perplexity** quantifies the uniformity of token usage frequency distribution across a codebook. Higher perplexity signifies a more uniform distribution, indicating balanced token utilization. It is calculated using the formula:
$$\text{Perplexity} = \exp H_v,$$
where $H_v$ represents as the entropy of the corpus for a given codebook vocabulary $v$. This is calculated as the sum of token entropy. To mitigate the impact of token length variability, the entropy is normalized by the average token length, and the adjusted entropy formula is given by:
$$H_v = -\frac{1}{l_v} \sum_{j \in v} P(j) \log P(j),$$
where $P(j)$ is the token usage frequency of token $j$ from the training corpus and $l_v$ is the average length of tokens in vocabulary $v$.

- **MUV** (Marginal Utility of Vocabularization) examines the benefits (entropy) a corpus can get from an increase of cost (size). A higher MUV indicates a more favorable benefit-cost ratio. This metric helps in understanding how efficiently the vocabulary expansion contributes to the overall utility of the codebook. It's defined as the negative derivation of entropy to size, as in the formula:
$$M_{v(k+m)} = \frac{-(H_{v(k+m)} - H_{v(k)})}{m}.$$
Here $v(k)$, $v(k+m)$ refer to vocabularies containing $k$ and $k+m$ tokens, respectively. $H_v$ denotes the entropy of the corpus with a given codebook vocabulary $v$ as defined above.

## C. Relevant Discussions

Quantizing protein structures in VQ-VAE-based PST methods addresses practical challenges and offers benefits. Some aspects are as follows:

- **Handling symmetry and physical constraints:** Protein structures are inherently redundant due to their trans-rotational equivariance and polymer nature. This SE-(3) requirements for protein structures have led to the development of invariant encoders and equivariant decoders that are computationally intensive and complex. Quantization simplifies these complexities by eliminating the need to explicitly model such constraints, thereby allowing models to focus on the most essential information needed for accurate protein structure representation.

- **Biological understanding:** Proteins often exhibit modularity, with distinct substructure motifs determining structure orientations and distinct domains responsible for different functions (Pearl et al., 2003). By discretizing protein structures into discretized tokens that reflect these functional units or motifs and domains, we gain a clearer insight into how each component contributes to the overall structure and function of the protein.

- **Preventing overfitting:** Employing discrete tokens instead of continuous features like coordinates and dihedral angles helps to reduce the risk of overfitting in modeling (Li et al., 2024). By transforming structure generation tasks into classification problems, rather than complex regression models, discrete tokens streamline the learning process and enhance model performance and generalizability.

- **Integration with large multimodal models:** Discrete structural and sequence tokens, along with text data, can be seamlessly integrated, facilitating the development of advanced multimodal large language models (LLMs). This also enables the application of optimization techniques developed in the natural language processing community for protein modeling.

## D. Benchmark Details

### D.1. Dynamic Programming Alignment Algorithm

To assess the similarity of the extracted structural tokens, we utilized Biotite (Kunzmann & Hamacher, 2018)'s "align_optimal()" function, which implements global alignment algorithms for pairwise sequence alignments using dynamic programming. The resultant alignment similarity is then used as the similarity for the structural tokens.

This versatile function supports both global alignments using the Needleman-Wunsch algorithm (Likic, 2008) and local alignments via the Smith-Waterman algorithm (Ligowski & Rudnicki, 2009). It can align two "Sequence" objects with potentially different alphabets, requiring a "SubstitutionMatrix" object that contains two alphabets of lengths $n$ or $m$, respectively, along with a similarity score matrix of

shape $(n, m)$. This flexibility to handle different alphabets significantly enhances the function's utility.

For discrete structural tokens, both the two alphabets were derived from a learned codebook token vocabulary, and "SubstitutionMatrix" was defined using scaled cosine similarities between codebook vectors $\boldsymbol{Q} = \{\boldsymbol{q}_i\}_{i=1}^K$:

$$\text{sim(i,j)} = \lfloor 100 * \cos(\boldsymbol{q}_i, \boldsymbol{q}_j) \rfloor, 1 \leq i, j \leq K.$$

For continuous structural tokens, each token was considered as an element in the alphabet, creating two alphabets sized according to the lengths $L_1$ and $L_2$ of the input protein structures. The similarity measure in the "SubstitutionMatrix" was computed similarly using the continuous latent representations $\{\boldsymbol{z}_i\}_{i=1}^{L_1}$ and $\{\hat{\boldsymbol{z}}_j\}_{j=1}^{L_2}$:

$$\text{sim(i,j)} = \lfloor 100 * \cos(\boldsymbol{z}_i, \hat{\boldsymbol{z}}_j) \rfloor, 1 \leq i \leq L_1, 1 \leq j \leq L_2.$$

# E. Method Details

## E.1. Overall Pipeline

AminoAseed is a VQ-VAE-based PST method, which takes protein frames as inputs for encoding, vector quantization, and decoding to reconstruct the protein structures.

**Protein frames as inputs.** Protein backbone structure is represented by the relative distance and orientation of frames defined by each residue's backbone coordinates. For residue $i$, its frame $\boldsymbol{T}_i \in SE(3)$ consists of a rotation matrix $\boldsymbol{R}_i \in SO(3)$ and a translation vector $\boldsymbol{t}_i \in \mathbb{R}^3$. The frame $\boldsymbol{T}_i$ can be calculated using the standard Gram-Schmidt algorithm (see Sec. E.4).

**Structure encoding.** For each residue $i$, its local neighborhood substructure obtains the 16 nearest residues (measured by $C_\alpha$ distance). The structure encoder input includes the frame for each residue, the frames for its neighboring residues, and their relative positional encodings. The structure encoder consists of a stack of geometric attention blocks, where each block contains a geometric self-attention layer (detailed in Sec. E.2) and a feedforward network (MLP+SwiGLU (Shazeer, 2020)). A linear layer is attached after the encoder to transform the hidden dimension of the encoder output to the codebook dimension before emplying vector quantization.

**Quantization.** The vector quantization process of AminoAseed is introduced in Sec. 4.2, while that of VanillaVQ is described in Sec. 2.2. The quantization process is not differentiable. To enable the gradient flow back to the encoder, the straight-through gradient estimation is applied and details can be found in Sec. E.3.

**Structure decoding.** After the quantization, the discrete latent representations are first transformed to the decoder

hidden dimension using a linear layer, then fed into the structure decoder. This decoder is composed using a stack of bidirectional transformer blocks with standard self-attention (Vaswani, 2017).

**Overall objective.** The overall training objective is described in Sec. 2.2, which includes the commitment term, quantization term, and the reconstruction term. In AminoAseed, the reconstruction term is further designed as the average of five structure reconstruction losses from ESM3: (1) **Geometric distance and direction losses** guarantee accurate backbone structure reconstruction; (2) **binned distance and direction classification losses** enhance early training convergence; and (3) **an inverse folding token prediction loss**, a cross entropy measure between the predicted and true sequences, serves as an auxiliary loss to enrich sequence-related information in the learned representations.

## E.2. Geometric Self-Attention Layer

The geometric self-attention layer (Hayes et al., 2025) involves transforming local frames into a global state to deploy a specialized attention mechanism, which assesses both rotational and distance similarities. After computing the attention scores and deriving the outputs weighted by these scores, the data is converted back to local frames. This ensures the model accurately captures the orientations and positional interactions among per-residue frames, essential for understanding the complex spatial relationships inherent in protein structures.

## E.3. Straight Through Estimator

As introduced in Sec. 2.2, the VQ layer maps the continuous latent representations $z$ into a discrete embedding $q_k$ using the codebook $M$. Because the discretization function is not continuously differentiable, a common approach for optimizing this layer is via a "straight-through estimator" (Bengio et al., 2013). This approach effectively sidesteps the non-differentiable nature of the function, permitting gradient updates through backpropagation to optimize the layer.

Specifically, before applying STE, we can analyze the process of backward gradient propagation in 3 distinct stages:

$$\frac{\partial \mathcal{L}}{\partial x} = \frac{\partial \mathcal{L}}{\partial q_k} \frac{\partial q_k}{\partial z} \frac{\partial z}{\partial x}$$

where $\frac{\partial \mathcal{L}}{\partial q_k}$ is the gradient through the decoder, $\frac{\partial q_k}{\partial z}$ is the gradient through the VQ layer, and $\frac{\partial z}{\partial x}$ is the gradient through the encoder. The non-continuous nature of the VQ transformation means $\frac{\partial q_k}{\partial z}$ is not computable.

To solve this non-differentiability issue, STE directly copies the gradients from $q_k$ to $z$, by modifying the input to the decoder to be $q_k - sg(z) + z$ instead of $q_k$. This modification

circumvents the vector quantization step for gradient computation. Simplifying this in the backpropagation formula, STE treats $\frac{\partial q_k}{\partial z}$ as the identity matrix $I$:

$$\frac{\partial \mathcal{L}}{\partial x} = \frac{\partial \mathcal{L}}{\partial q_k} I \frac{\partial z}{\partial x}.$$

This alteration ensures that the entire backward pass remains differentiable despite the non-differentiability of VQ layer.

### E.4. Gram-Schmidt Algorithm

The Gram-Schmidt algorithm (Björck, 1994) is a method of constructing an orthonormal basis from a set of vectors in an inner product space. In the algorithm, a translation vector $\overline{t}$, and two vectors $\overline{x}$ and $\overline{y}$ define the local $x$-$y$ plane, as illustrated in Alg. 1.

---
**Algorithm 1** Gram-Schmidt Process
---
**Require:** $\overline{t} \in \mathbb{R}^{L \times 3}, \overline{x} \in \mathbb{R}^{L \times 3}, \overline{y} \in \mathbb{R}^{L \times 3}$
1:  $\hat{x} = \frac{\overline{x}}{\|\overline{x}\|}$
2:  $e_1 = \overline{y} - (\hat{x}^\top \overline{y})\hat{x}$
3:  $\hat{e}_1 = \frac{e_1}{\|e_1\|}$
4:  $e_2 = \hat{x} \times \hat{e}_1$
5:  $R = [\hat{x}, \hat{e}_1, e_2]$
6:  $T = \begin{bmatrix} R & t \\ 0_{1 \times 3} & 1 \end{bmatrix}$
7:  return $T$

---

To create the residue frames $T_i$, we follow ESM3 (Hayes et al., 2025) to apply $C_\alpha$ at the origin of the frame, $C$ on the negative x-axis $(-\overline{x})$, and $N$ on the xy-plane.

The resulted frame $T_i$ can be represented as

$$T_i = \begin{pmatrix} R_i & t_i \\ 0_{1 \times 3} & 1 \end{pmatrix} \in SE(3)$$

The rotation matrix $R_i \in SO(3)$ rotates vectors to a local coordinate system where the $N$-$C_\alpha$-$C$ plane for the corresponding residue spans the $x$-$y$ planel; and the translation vector $t_i \in \mathbb{R}^3$ specifies the position of the residue's $C_\alpha$.

In the end, $R_i$ is composed of three 3-dimensional vectors $[\hat{e}_1, \hat{e}_2, \hat{e}_3]$, where $\hat{e}_1$ and $\hat{e}_2$ are orthogonal unit vectors on the $N$-$C_\alpha$-$C$ plane, and $\hat{e}_3$ is a unit vector perpendicular to the plane.

## F. Experimental Result Details

### F.1. Setup

**Pre-training dataset preparation details.** We applied the same criteria as those used for training the OpenFold2 model: (1) PDB structures deposited before 2020-05-01; (2) resolution better than or equal to 9Å; (3) protein chain length greater than 20; (4) no more than 20% of the sequence is the same amino acid. We then downsampled 10% of the protein chains to train our PSTs, resulting in 48,316 chains. The down-sampling is supported by the fact that training a protein folding model with as few as 1,000 protein chains achieved a decent performance (Ahdritz et al., 2024). The downsampled protein chains have lower than 40% sequence identity to each other. To reduce the memory cost, we filtered out proteins longer than 512.

**Pre-training configuration details.** We employed the same configuration for both of our implemented models AminoAseed and VanillaVQ. Specifically, our models were trained using an Adam optimizer with a linear warmup schedule to a peak learning rate of 0.0001, followed by cosine decay to 10% of the peak learning rate. We use a weight decay of 0.01. The training process involved 5,426 warmup steps and continued for a total of 108,530 steps, lasting approximately 30 hours on 8 NVIDIA A100 GPUs. Each GPU processed a batch size of 4, without gradient accumulation, resulting in an effective global batch size of 32. During pre-training, DeepSpeed ZeRO training stage 2 (Rajbhandari et al., 2020) was employed to reduce GPU memory footprint and enhance the full precision training of our models, by sharding optimizer states and gradients across GPUs.

For the VQ-VAE architecture, the codebook size was set to 512 with each codebook vector having a dimension of 1024, matching the capacity of ESM3's structure tokenizer to ensure a fair comparison. The encoder utilized 2 geometric attention blocks, featuring a hidden dimension size of 1024 and 128 geometric attention heads. The decoder employed 8 traditional bi-directional self-attention blocks, each with a hidden dimension of 1024 and 16 attention heads. Additionally, the loss weight $\beta$ for the commitment loss was fixed at a constant of 0.25.

**Training and evaluation configuration details for supervised downstream tasks.** For supervised tasks, the continuous or discrete structural representations extracted by PSTs, along with absolute positional encodings were fed into a LayerNorm layer and dropout layer with a dropout ratio of 0.1, before proceeding to the probing layer. And the probing layer consisted of a two-layer MLP with a hidden dimension of 512, ReLU nonlinearity, and a dropout layer with a dropout ratio of 0.1 between the layers. This layer produced per-residue logits for functional site prediction (binary classification prediction), regression scores for physicochemical property tasks (regression prediction), and per-protein logits for structural property prediction (multi-class classification).

Training was managed using an Adam optimizer with a cosine annealed learning rate schedule, selecting peak learning rates from the set $\{0.1, 0.01, 0.001, 1e{-}4, 5e{-}5, 1e{-}5, 5e{-}5, 1e{-}6\}$. The best learning rate was chosen based on the

best validation Macro F1 for classification tasks, and best validation Spearman's $\rho$ for regression tasks. The training protocol included 200 warmup steps and a total of 10,000 training steps. Each experiment was conducted on a single NVIDIA A10 GPU, with a per-GPU batch size of 8 for all supervised tasks, except for the Homo task, which used a batch size of 64. To manage peak memory usage effectively, protein sequences exceeding 600 residues were filtered out. All results reported were obtained using seed 1,234.

**Ablation experiment configurations for using continuous structural representations.** For VQ-VAE-based PSTs, we used their encoder output as the corresponding continuous counterparts, substituting them with the discrete structural representations as the input for supervised tasks. All other training and evaluation settings were kept the same as those in "Downstream Effectiveness" benchmarking for a fair comparison. Notably, ProTokens was excluded because its released implementation did not grant access to its encoder output.

**Ablation experiment configurations for combining amino acids with structural representations.** We applied a learnable embedding layer to the 20 amino acid types, and add their embedding to the structural representations before entering into the LayerNorm and dropout layer, before proceeding into the probing layer. All other training and evaluation settings were kept the same as those in "Downstream Effectiveness" benchmarking for a fair comparison.

**Ablation experiment configurations for adding noise to structural representations.** We masked the structural representations residue-wise and replaced them with a learnable [MASK] embedding to simulate adding noise. We calculated the proportion of masked residues in a protein as the noise level, as shown in Fig. 5 and Fig. 11. Intuitively, a higher noise level indicates that the structural representations contain less meaningful information. All other training and evaluation settings were kept the same as those in "Downstream Effectiveness" benchmarking for a fair comparison.

**Scaling experiment configurations for varying codebook sizes.** We fixed the size of the codebook matrix $K \times D$ at $4096 \times 128$, and varied the codebook size $K$ and dimension $D$. The codebook size $K$ ranged from $2^6 = 64$ (with $D = 8192$) to $2^{19}$ (with $D = 1$). We also fixed the sizes of encoder and decoders across codebook sizes, such that there were only a small differences in encoder and decoder sizes across different codebooks, due to a linear mapping to align the encoder output hidden dimension and decoder input hidden dimension with the codebook dimension.

We pretrained these models on our pre-training data for three different compute budgets ($\{1e{+}18, 3e{+}18, \text{ and } 1e{+}19\}$ floating point operations), with number of warm-up steps set to 5% of total steps. The peak learning rate was established

Table 9: Model comparison.

| Model | #Params | Model Input |
|---|---|---|
| ProteinMPNN | 1.7M | Backbone Structure |
| MIF | 3.4M | Backbone Structure |
| FoldSeek | 282 | Backbone Structure |
| ProTokens | 34M | Backbone Structure |
| AIDO.st | 275M | Backbone Structure |
| ESM3 | 30M | Backbone Structure |
| VanillaVQ | 31M | Backbone Structure |
| AminoAseed | 31M | Backbone Structure |
| Cheap | 96M | All-atom Structure & Sequence |

at $1e-4$, utilizing a linear warm-up followed by cosine decay learning rate schedule to 10% of the peak learning rate. Global batch size was set to 32 proteins.

**Scaling experiment configurations for varying encoder sizes.** We fixed the size of the codebook matrix $K \times D$ at $4096 \times 128$ and the decoder of 8 traditional self-attention blocks, and then varied the size of the encoders by changing the hidden dimention sizes (from 32 to 2048) and number of geometric attention layers (from 1 to 4) while fixing the number of geometric attention heads to 128. The total number of encoder parameters ranges from 107K to 96M.

We pretrained these models on our pre-training data under five different compute budgets: $\{3e{+}17, 1e{+}18, 3e{+}18, 1e{+}19, \text{ and } 3e{+}19\}$ floating point operations. The training configurations were the same as the scaling experiments that varying the size of the codebook described above.

**Baseline model size comparison.** The number of parameters for each PSTs is summarized in Tab. 9. FoldSeek is very lightweight. Top-performaing VQ-VAE-based PSTs have about 30M parameters, while IF-based PSTs contain approximately 2M parameters.

### F.2. More Sensitivity Results

**Using TM-score instead of the negative RMSD as structural similarity.** We also used the negative value of RMSD as a structural similarity measure, and summarized the results in Tab. 10. Compared to using TM-score in Tab. 5.3, the correlation values for RMSD presented across all methods are significantly lower. This is because RMSD is sensitive to outliers and local structural discrepancies, while TM-score focuses on global structure and is less sensitive to local structure variations. This comparison proves TM-score more suitable for assessing structural similarity in our StructTokenBench for sensitivity perspective evaluation. Despite RMSD's challenges, AminoAseed significantly outperforms other models, suggesting AminoAseed's superiority in detecting structure conformational changes.

### F.3. More Ablation Study Results

**Remaining task splits for performance comparisons between discrete and continuous representations under**

Table 10: Sensitivity evaluation on conformational proteins, with conformer structural similarity measured by the negative RMSD instead of the TM-score used in Tab. 3.

| **Model** | Apo Holo | | Fold Switching | |
|---|---|---|---|---|
| | PCC% | Spearman's $\rho$% | PCC% | Spearman's $\rho$% |
| ProteinMPNN | -0.003 | 0.0643 | 0.0057 | 0.0504 |
| MIF | 0.0033 | 0.0517 | 0.0442 | 0.0878 |
| FoldSeek | -0.0112 | 0.0407 | 0.0073 | 0.0475 |
| ProTokens | 0.0807 | 0.1569 | 0.1022 | 0.1161 |
| ESM3 | -0.0442 | 0.133 | 0.0944 | 0.1252 |
| VanillaVQ | 0.0479 | 0.1247 | 0.0715 | 0.1080 |
| AminoAseed | **0.1207** | **0.2076** | **0.2737** | **0.3172** |

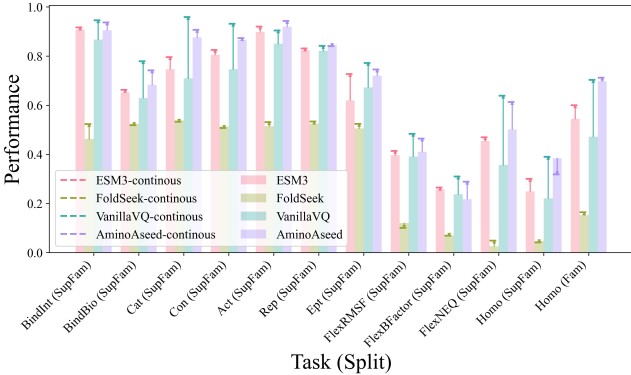

Figure 10: Performance comparison of using continuous versus discrete structural representations for the remaining supervised task splits, in complement to Fig. 4.

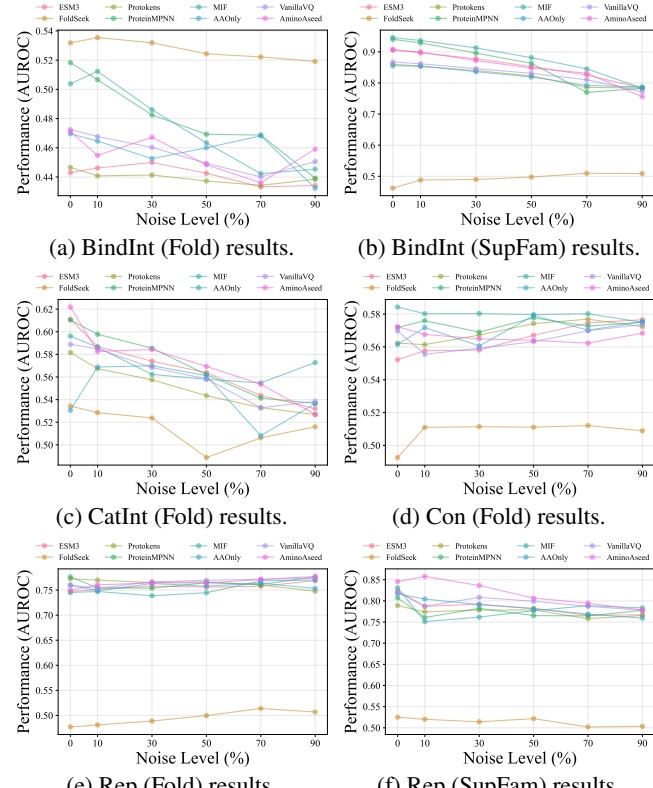

(a) BindInt (Fold) results.    (b) BindInt (SupFam) results.

(c) CatInt (Fold) results.    (d) Con (Fold) results.

(e) Rep (Fold) results.    (f) Rep (SupFam) results.

Figure 11: Supervised performance with increasing noises in the PST-extracted structural representations, with more task splits reported in complement to Fig. 5.

**supervised downstream tasks.** As shown in Fig. 10, for most SupFam task splits, continuous representations outperform discrete ones, although the gains is marginal for ESM3, FoldSeek, and AminoAseed. This observation, consistent with the finding in Sec. 5.6, indicates that the continuous format is not the primary factor contributing to performance enhancement.

Notably, the continuous representations from VanillaVQ match or exceed those from AminoAseed, bridging the performance gap observed with their discrete counterparts. This suggests that while the encoder of VanillaVQ could be optimized similarly to AminoAseed, AminoAseed optimizes its codebook more effectively using its proposed engineering techniques, thereby better aligning the encoder output with the codebook vectors with reduced distribution shift.

**Extended supervised task split results for adding noises to the structural representation.** In addition to the previously reported Con (SupFam) and CatInt (SupFam) in Fig. 5, we include results for Con (Fold) and CatInt (Fold), as well as two additional tasks, BindInt and Rep, across both Fold and SupFam splits in Fig. 11. For most of these task splits, AminoAseed consistently outperforms ESM3 across various noise levels.

### F.4. More Scaling Study Results

**Scaling of codebook sizes.** We found VanillaVQ is not effective in minimizing the quantization loss, leading to up to 2 magnitudes higher quantization and overall loss across codebook sizes (Fig. 12), highlighting the benefit of our reparameterization approach in effectively learning the codebook. We also observed that AminoAseed consistently achieved better reconstruction qualities over VanillaVQ across compute budgets and codebook sizes (Fig. 6).

When analyzing the PST qualities across codebook sizes and compute budgets (Fig. 6), we noticed that: (1) reconstruction qualities turn to be more similar across codebook sizes as compute increase; (2) the trend in UR and downstream effectiveness tasks such as CatInt maintained across compute budgets. These observations suggest that although reconstruction quality can be improved with more compute regardless of the codebook sizes used, one cannot compensate the low UR and downstream effectiveness from suboptimal codebook sizes with more compute.

We also note that although large codebook sizes $2^{11}$ achieved optimal reconstruction quality, it suffers from low UR and downstream effectiveness (Fig. 6, Fig. 12). This hints a trade-off between structure generation and token

effectiveness: larger codebook sizes may help the decoder to generate protein structures more accurately, it limits the downstream effectiveness of the structure tokens for supervised downstream tasks and reduces utilization rates.

We highlight that the large codebook size also pose additional challenges: (1) it is hard to analyze all the structure patterns; (2) it leads to difficulty in downstream predictive tasks, as the predictive space is large and similar code vectors could confuse each other.

**Scaling of encoder sizes.** We noted that the encoders in VQ-VAE-based PSTs (both AminoAseed and VanillaVQ), do not demonstrate power-law scaling with compute and training data ( Fig. 7). We speculate this is likely due to (1) the inherent redundancy in protein structure data; and (2) the performance is limimted by the decoder capacity. Future research is needed to delve deeper into the scaling laws of VQ-VAE models.

### F.5. More Visualization Results

**More visualization of local neighborhoods where protein conformer structure variations are correctly detected.** In Fig. 13 we show three common examples of AminoAseed's "vocabulary". In each case, we show an example from three different proteins, co-aligned using the residue with the indicated token index, and the residues just before and just after that in the sequence. A very common example (Fig. 13(a)) is a simple turn linking other secondary structural elements. The second example (Fig. 13(b)) is an alpha helical element, although we note that while the backbones align almost perfectly, conventional tools for detecting alpha helices do not agree about whether these three examples are part of an alpha helix or not. Finally, we show an example of one $\beta$-sheet token (Fig. 13(c)).

Another way to visualize the differences between tokenizers is to see how the token indices change when there are structural changes in a particular protein, as there are when binding a small molecule. We can compare the *apo* (without ligand) and *holo* states to look for differences in the tokenization. In Fig. 14 we compare the AminoAseed tokenizer to ESM3's structure tokenizer. To illustrate, we compare the apo (PDB 1LIP) and holo (PDB 1JTB) states of a lipid transfer protein. The upper left panel of Fig. 14 shows that to accommodate the ligand, an $\alpha$-helix displaces to the right, as the loop in the foreground extends. AminoAseed differs from ESM3 in where exactly this extension from apo to holo becomes detectable, highlighting residues 61 and 64, while ESM3 highlights residues 62 and 63. There is not a clear winner here: the rest of Fig. 14 shows the structure aligned at residue 64 (upper right), 63 (lower left) and 61 (lower right). The backbone environment around each residue appears to be different between apo and holo forms, yet neither tokenizer distinguishes all positions. AminoAseed detects

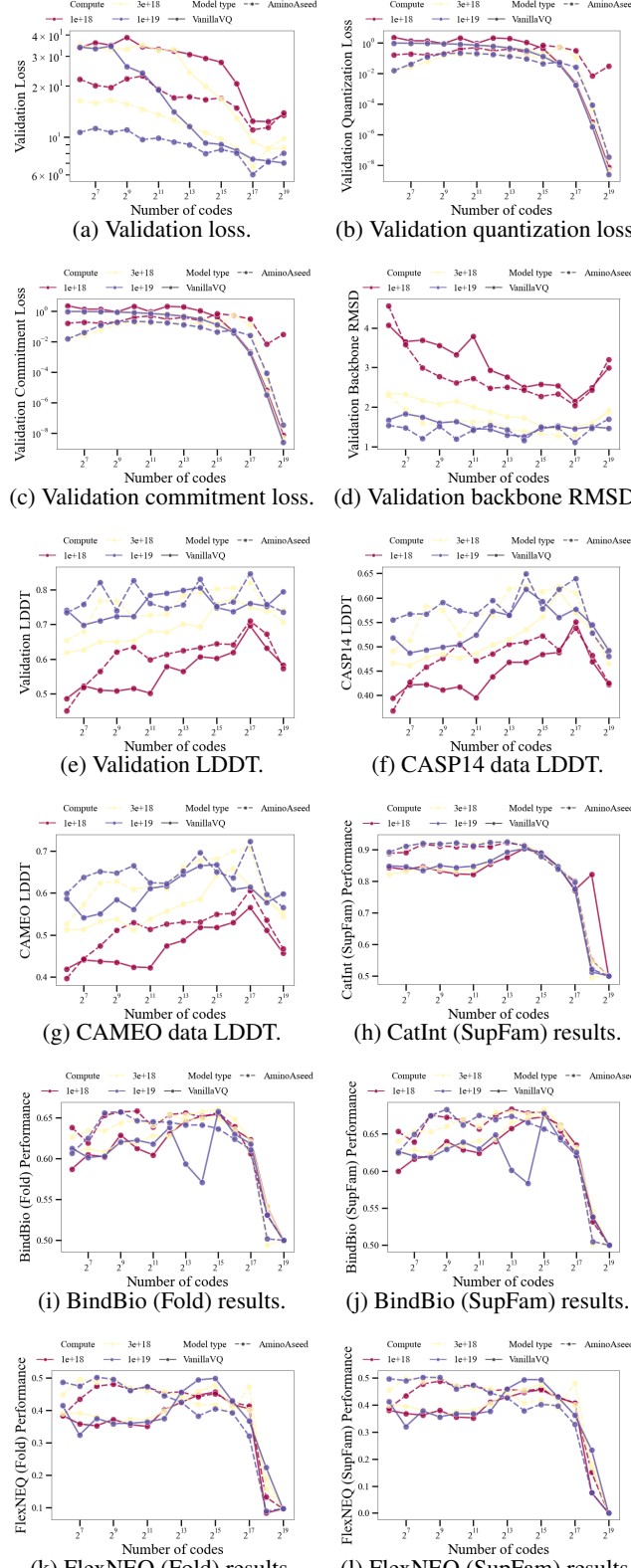

(a) Validation loss.

(b) Validation quantization loss.

(c) Validation commitment loss.

(d) Validation backbone RMSD.

(e) Validation LDDT.

(f) CASP14 data LDDT.

(g) CAMEO data LDDT.

(h) CatInt (SupFam) results.

(i) BindBio (Fold) results.

(j) BindBio (SupFam) results.

(k) FlexNEQ (Fold) results.

(l) FlexNEQ (SupFam) results.

Figure 12: Scaling impact of codebook sizes on: (a-c) different losses on holdout validation set; (d-g) reconstruction quality on validation set, CASP14 and CAMEO test set; (h-l) more supervised task split performance evaluating the Downstream Effectiveness perspective of PSTs.

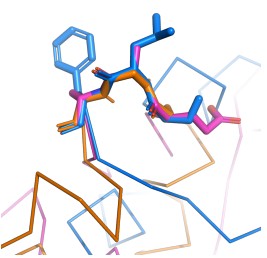

(a) Token ID 358 for PDB IDs 7ABW Leu232 (marine blue), 7M7A His307 (orange), 7MHU Ala214 (magenta) display a common turn motif.

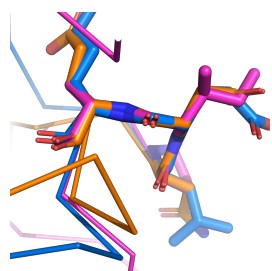

(b) Token 47 for PDB IDs 7MHU Leu287 (magenta), 7M7A Glu465 (marine blue), 6POO Asp258 (orange) display an alpha-helical motif.

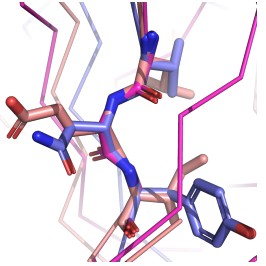

(c) Token 253 for PDB IDs 7MHU Ala69 (magenta), 7K7W Glu74 (salmon), 6S44 Asn80 (blue) display a common $\beta$-sheet motif.

Figure 13: Examples of our structural vocabulary. Motifs are shown with side chains visible, the remainder of each protein is shown only as a $C_\alpha$ ribbon.

the most subtle difference, at position 64, but misses a much less subtle difference at positions 62-63. Both tokenizers agree that the apo and holo forms of remainder of this loop are different at each position.

**t-SNE for discrete codebook vectors and continuous structural representations from PST encoders.** In Fig. 15(a), on the one hand, ProTokens, ESM3, and AminoAseed display distinct patterns in their codebooks for per-residue protein structural representations. ProTokens' representations form many small, tightly clustered groups that are widely spaced from each other. In contrast, ESM3 and AminoAseed exhibit more diverse distributions, featuring both large and small clusters. On the other hand, the codebook vectors of FoldSeek and VanillaVQ appear more normally distributed. This is likely because FoldSeek has too few codes, while VanillaVQ starts with a random initialization from a normal distribution, with only a small number

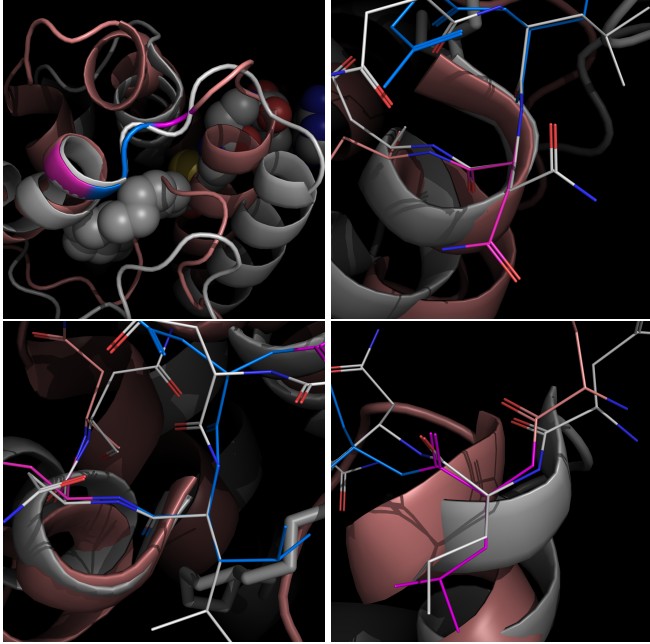

Figure 14: *apo* (salmon) and *holo* (grey) structures of a lipid transfer protein illustrate tokenizer differences. **Upper left** residues tokenized differently in apo and holo states by AminoAseed are shown in magenta; those tokenized differently by ESM3 are shown in blue; others are tokenized differently by both models. **Upper right** Aligned on the backbone of residue 64. **Lower left** Aligned on residue 63. **Lower right** Aligned on residue 61.

of codes optimized, which is visible in the upper right of VanillaVQ's distribution plot and appears as a deviation from the initial setup. Moreover, the comparison between VanillaVQ and AminoAseed highlights the effectiveness of our proposed engineering techniques in enhancing codebook optimization.

In Fig. 15(b) and (c), we present the continuous structural representations extracted from the PST's encoder, applied to the CASP14 and CAMEO datasets, respectively. The encoders for ESM3, VanillaVQ, and AminoAseed have successfully learned discernible patterns.

### F.6. More Baselines

As suggested by anonymous reviewers, we add two more baselines: AIDO.st (Zhang et al., 2024d) which is a VQ-VAE-based PST, and Cheap (Lu et al., 2024) which replaces the quantization step with a tanh layer to get continuous embeddings instead of discretized embeddings used in VQ-VAE. Specifically, AIDO.st takes backbone structures as input and is thus directly comparable to all results in the main text; while Cheap models all-atom structures and protein sequences as input, placing it outside this benchmark's

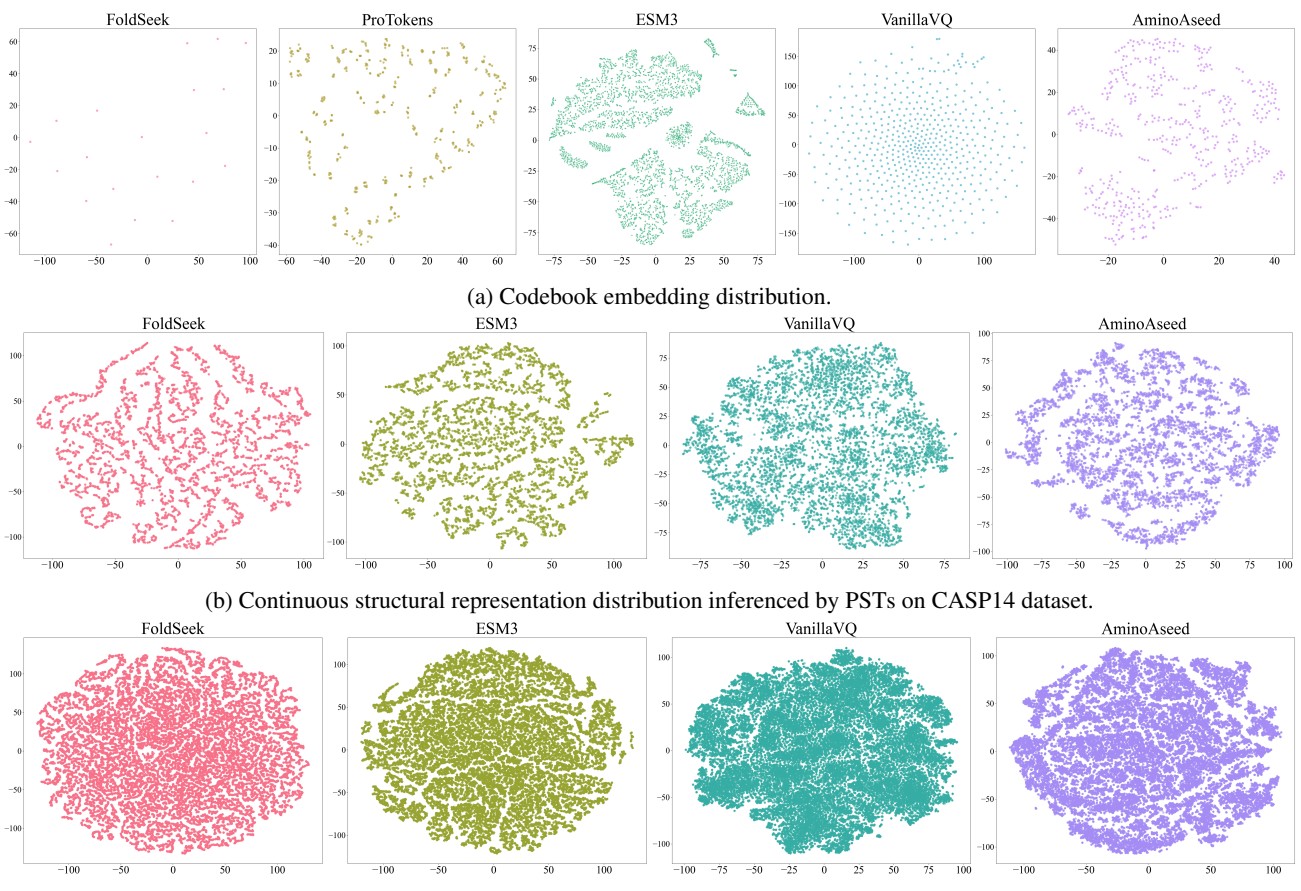

(a) Codebook embedding distribution.

(b) Continuous structural representation distribution inferenced by PSTs on CASP14 dataset.

(c) Continuous structural representation distribution inferenced by PSTs on CAMEO dataset.

Figure 15: Visualization for discrete and continuous structural representations extracted from PSTs using t-SNE.

primary focus. Their model sizes are summarized in Tab. 9.

As shown in Tab. 11, Tab. 12 and Tab. 13, despite its large size (275M parameters), AIDO.st is less effective than AminoAseed on downstream effectiveness, and also fall short in sensitivity. However, AIDO.st achieves a very high codebook utilization efficiency. Cheap beats IF-based PSTs and AminoAseed on supervised tasks, but largely lags behind AminoAseed in sensitivity. Codebook utilization efficiency result is not applicable for Cheap as it has no codebook.

## G. Related Work Details

**VQ-VAE-based PSTs have been rapidly emerging.** Fold-Seek (Van Kempen et al., 2024), the pioneering PST model, significantly speeds up protein structure alignment. Subsequent models targets on enhancing protein language models with structure knowledge, either building on Foldseek's tokenization, such as ProstT5 (Heinzinger et al., 2023) and SaProt (Su et al., 2023), or building their own tokenization model, like ProSST (Li et al., 2024) and ESM3 (Hayes et al., 2025). ProTokens (Lin et al., 2023a) facilitates protein structure generation by leveraging the quantized codes to remove group symmetry and polymer restraints. To enhance the quality of PSTs, some methods refine the standard VQ-VAE training. Notably, Gaujac et al. (2024) applies Finite Scalar Quantization (FSQ) (Mentzer et al., 2023), and Fold-Token (Gao et al., 2024) employs Soft Conditional Vector Quantization. Other approaches focus on domain-specific enhancements: Bio2Tokens (Liu et al., 2024) expands the structural resolution from backbone atoms to all-atom level, while FoldToken (Gao et al., 2024) and CHEAP (Lu et al., 2024) propose to tokenize the joint distribution of sequences and structures, instead of just structures, to better capture the complex interplay between the two modalities.

Table 11: Benchmark results for supervised downstream tasks for two additional baselines. The table format follows Tab. 2.

| Task | Split | Model | |
|---|---|---|---|
| | | AIDO.st | Cheap |
| **Functional Site Prediction (AUROC%)** | | | |
| BindInt | Fold | 44.66 | 59.87 |
| | SupFam | 84.21 | 97.38 |
| BindBio | Fold | 65.50 | 85.59 |
| | SupFam | 66.70 | 88.27 |
| BindShake | Org | 69.28 | 87.74 |
| CatInt | Fold | 57.30 | 65.20 |
| | SupFam | 81.94 | 97.14 |
| CatBio | Fold | 73.72 | 93.91 |
| | SupFam | 78.66 | 95.78 |
| Con | Fold | 56.64 | 61.12 |
| | SupFam | 73.79 | 95.14 |
| Rep | Fold | 77.69 | 77.35 |
| | SupFam | 78.08 | 80.45 |
| Ept | Fold | 60.26 | 64.13 |
| | SupFam | 72.30 | 78.83 |
| **Average** AUROC% | | 69.38 | 81.86 |
| **Physicochemical Property Prediction (Spearman's $\rho$%)** | | | |
| FlexRMSF | Fold | 33.15 | 50.56 |
| | SupFam | 26.93 | 48.59 |
| FlexBFactor | Fold | 18.88 | 37.17 |
| | SupFam | 19.31 | 37.90 |
| FlexNEQ | Fold | 16.41 | 60.69 |
| | SupFam | 16.17 | 60.18 |
| **Average** $\rho$% | | 21.81 | 49.18 |
| **Structure Property Prediction (Macro F1%)** | | | |
| Homo | Fold | 7.89 | 40.55 |
| | SupFam | 4.77 | 58.50 |
| | Fam | 13.71 | 92.89 |
| **Average** Macro F1% | | 8.79 | 63.98 |

Table 12: Sensitivity evaluation on conformational proteins for two additional baselines.

| Model | Apo Holo | | Fold Switching | |
|---|---|---|---|---|
| | PCC% | Spearman's $\rho$% | PCC% | Spearman's $\rho$% |
| AIDO.st | 43.02 | 54.25 | 61.59 | 66.11 |
| Cheap | 33.38 | 40.20 | 48.34 | 52.51 |

Table 13: Codebook utilization efficiency evaluation on CASP14 and CAMEO datasets for AIDO.st. Note that Cheap does not contain discrete codebooks, thus cannot be evaluated for efficiency.

| Model | #Code ($K$) | Dim ($D$) | CASP14 | | | CAMEO | | |
|---|---|---|---|---|---|---|---|---|
| | | | UR% | Perplexity | MUV | UR% | Perplexity | MUV |
| AIDO.st | 512 | 384 | 88.05 | 0.7729 | 2.52e-4 | 95.12 | 0.8266 | 2.50e-4 |

