# OpenReview forum: "Protein Structure Tokenization: Benchmarking and New Recipe"
_ICML.cc/2025/Conference — ICML 2025 poster_

### Official Review · Reviewer_NnWf · 2025-02-17

**Overall Recommendation:** 4

**Summary:**

This study considers the problem of protein structure tokenization, which it defines as the problem of distilling protein 3D structure into discrete or continuous representations. The authors introduce two main contributions: 1) a set of benchmark datasets for evaluating protein structure tokenization models and algorithms and 2) AminoAseed, a new method for protein structure tokenization that leverages codebook gradient updates.

**Claims And Evidence:**

Most of the claims made throughout the paper are well supported by the evidence provided. The explanation about the datasets used is convicing and the results demonstrating that their new method outperforms the state-of-the-art is also convincing. The ablation study is both complete and convincing.

The paper benchmarks two distinct qualities Distinctiveness of the vectors in the codebooks and Efficiency of the codebook utilisation. Intuitively, these two values should be related to each other. A more distinct codebook should lead to more efficient utilisation. However, the empirical results show the opposite: Foldseek, which has the worst curve shape for the distinctiveness evaluation, has much better performance on the Efficiency benchmark. This discrepancy suggests to me that this is some problem in how either efficiency or distinctiveness are measured.

I would argue that the claim in the abstract that ESM3 is "the leading model" is not supported by the evidence in the benchmark results provided where MIF is the leading model. Maybe this should be rephrased to "the leading VQ-VA-based model".

**Essential References Not Discussed:**

None that I've been able to identify.

**Experimental Designs Or Analyses:**

Yes, I think that the experimental design is sound. I've already discuss some ambiguous experimental descriptions that I think should be clarified. Otherwise, I think the experimental design is sound.

**Methods And Evaluation Criteria:**

One of the main focus of the paper is to define the datasets and evaluation criteria to benchmark structure tokenization methods. I think that the choices made by the authors make sense and well reasoned.

Benchmark construction
---
- Effectiveness datasets make emphasis on the detection of local substructures with functional relevance (binding site, catalytic, conserved site prediction), that greatly overlap with each other; and only one of global information (remote homology); however I generally agree with this choice as they are the closest tasks to real-world data.
- Sensitivity, the general idea makes a lot of sense and it is quite reasonable. It assumes that TM-score is the gold standard, which is not, it is the state-of-the-art structure search algorithm and I think an acknowledgement on this limitation would be useful. On the other hand, the target value of representation similarity is computed via dynamic programming, but it is not clearly specified whether it is global (or optimal) alignment or local (suboptimal). It should be clearly stated both in the main text and the appendix.
- Distinctiveness, I find this benchmark/metric the one with the weakest foundation, as it makes the assumption that cosine similarity is a good measurement of the distinctiveness of the vectors, which I do not think is properly justified. This might be the source of the discrepancy with efficiency
-  Efficiency, I think this evaluation makes sense.

New tokenization method
---
- The study of optimal codebook size and dimension is quite informative, the methodology is simple and sound.
- The finding of discretised representations obtaining similar performance to continuous ones is intriguing.
- The finding that structural representations are less robust than sequence representations is interesting and shows that there is still room for improvement in the field on how to combine these types of representations

**Other Comments Or Suggestions:**

I would like to congratulate the authors on a really interesting and solid study. I would update my recommendation to strong accept if my concerns are satisfactorily address. Particularly, the issue of the discrepancy between distinctiveness and efficiency.

**Other Strengths And Weaknesses:**

None, the main strengths and weaknesses have already been discussed in the prior sections

**Questions For Authors:**

- Do you have any thoughts/insights into the relationship of distinctiveness and efficiency? In my mind, they are related to each other, but I would like to know what your thoughts are on the topic and whether my intuition might be wrong.

**Relation To Broader Scientific Literature:**

The work studies Protein Structure Tokenization, which is a problem first tackled through the lens of deep learning, by [1]. It is also related to the problem of inverse folding which has been tackled by [2, 3], which is also related to protein design. It is also related to multimodal integration of protein sequence and structure [5, 6]. The benchmarking of protein representation techniques is related to [9 - 11].


1. van Kempen M, Kim SS, Tumescheit C, Mirdita M, Gilchrist CL, Söding J, Steinegger M. Foldseek: fast and accurate protein structure search. Biorxiv. 2022 Feb 9:2022-02.
2. Hsu C, Verkuil R, Liu J, Lin Z, Hie B, Sercu T, Lerer A, Rives A. Learning inverse folding from millions of predicted structures. InInternational conference on machine learning 2022 Jun 28 (pp. 8946-8970). PMLR.
3. Dauparas J, Anishchenko I, Bennett N, Bai H, Ragotte RJ, Milles LF, Wicky BI, Courbet A, de Haas RJ, Bethel N, Leung PJ. Robust deep learning–based protein sequence design using ProteinMPNN. Science. 2022 Oct 7;378(6615):49-56.
4. Yang, K. K., Zanichelli, N., and Yeh, H. Masked inverse folding with sequence transfer for protein representation learning. Protein Engineering, Design and Selection, 36:gzad015, 2023.
5. Heinzinger M, Weissenow K, Sanchez JG, Henkel A, Mirdita M, Steinegger M, Rost B. Bilingual language model for protein sequence and structure. bioRxiv. 2023 Jul 25:2023-07.
6. Su J, Han C, Zhou Y, Shan J, Zhou X, Yuan F. Saprot: Protein language modeling with structure-aware vocabulary. bioRxiv. 2023 Oct 2:2023-10.
7. Rao R, Bhattacharya N, Thomas N, Duan Y, Chen P, Canny J, Abbeel P, Song Y. Evaluating protein transfer learning with TAPE. Advances in neural information processing systems. 2019;32.
8. Capel H, Weiler R, Dijkstra M, Vleugels R, Bloem P, Feenstra KA. ProteinGLUE multi-task benchmark suite for self-supervised protein modeling. Scientific Reports. 2022 Sep 26;12(1):16047.
9. Kucera T, Oliver C, Chen D, Borgwardt K. ProteinShake: building datasets and benchmarks for deep learning on protein structures. Advances in Neural Information Processing Systems. 2024 Feb 13;36.
10. Zhang C, Zhang X, Freddolino PL, Zhang Y. BioLiP2: an updated structure database for biologically relevant ligand–protein interactions. Nucleic Acids Research. 2024 Jan 5;52(D1):D404-12.
11. Unsal S, Atas H, Albayrak M, Turhan K, Acar AC, Doğan T. Learning functional properties of proteins with language models. Nature Machine Intelligence. 2022 Mar;4(3):227-45.

**Theoretical Claims:**

There are no significant theoretical claims. The study is mostly interested in practical application of well established concepts.

---

> ### Author Rebuttal · Authors · 2025-04-01
>
> Thanks for your positive review and insightful comments! We respond to your concerns as below:
>
> > R1: Discrepancy observation that FoldSeek, which has the worst curve shape for the distinctiveness evaluation, has much better performance on efficiency benchmark. However, the distinctiveness and efficiency metrics should be related to each other. A more distinct codebook should lead to more efficiency utilization
>
> We would like to humbly argue that **the two distinctiveness and efficiency are related, but it is not necessarily the case that as one increases, and the other increases**.
> 1. Importantly, **the discrepancy stems from the different codebook size for different PST methods**. And the reviewer’s intuition would be more sound when the codebooks share the same size: a more distinct codebook should lead to more efficiency utilization. **For example**, utilization is measured by the factor of #used codes and #total codes (i.e., the codebook size). Smaller the codebook size, higher the chance for utilization rate to be higher. **FoldSeek has a very small codebook size K=20** (see Tab. 4), so it’s natural to have a high utilization rate, despite the observation that its used codes are not that distinct from each other.
> 2. We would like to clarify **the concept of distinctiveness**. The codebook has two parts of codes: used and unused ones. For distinctiveness, **among the utilized codes in a codebook**, the more distinct they are, **the better chance for them to avoid ambiguous token-substructure mappings**.  We would like to emphasize the distinctiveness with used codes, and we show it in Fig. 3 (right panel). Because in reality, those used codes matter the most.
>
> **We’re open to hear more opinions on this topic from the reviewer**, and are always eager to learn more to enrich our knowledge.
>
>
> > R2: Question on the meaning to have distinctiveness evaluation, given its discrepancy to “codebook utilization”.
>
> 1. The **“Distinctiveness” evaluation was motivated from the observation in ESM3’s** (Fig. S5) that ambiguous structural token mapping can hinder interpretability and harm model capability. **“Distinctiveness” measures the similarity within codebook vectors, which serves as a proxy to assess token ambiguity.**
> 2. We understand the reviewer’s concern that simply using cosine similarity for codebook vectors might be limited. **We would like to continue working on a better metric for measurement instead of simple cosine similarity. **
> 3. Please see **R1** for more discussion on the “discrepancy between distinctiveness metric and codebook utilization metric”. In general, we would like humbly argue that the discrepancy mainly stems from **the different codebook size used in different PST methods**, but not the metrics themselves. We’re always open to hear more opinions on this topic from the reviewer, as we’re also in the exploration stage to find a good way to benchmark PST methods.
>
>
> > R3: “ESM3 is the leading model” should be rephrased to “the leading VQVAE-based model”
>
> We agree and would modify this in paper.
>
> > R4: In sensitivity evaluation, TM-score is not a golden standard for structure similarity, but a state-of-the-art structure search algorithm. Acknowledgement of this limitation is useful.
>
> Thanks for the valuable suggestion. We would acknowledge this limitation in both main text and appendix that TM-score is not a golden standard to measure structure similarity.
>
> Also, commonly used metrics to measure structure similarity include TM-score (more globally) and RMSD (more locally). In App. F.2, we explored using RMSD instead of the TM-score in “Sensitivity” evaluation. As shown in Tab. 10, using RMSD is less effective than TM-scores. This might be because RMSD is sensitive to outliers and local structural discrepancies, while TM-score focuss on global structure and is less sensitive to local structure variations.
>
> > R5: It’s unclear whether the dynamic programming in sensitivity used global or local alignment, which should be stated both in main text and appendix
>
> We used the global alignment algorithm by adopting the Biotite’s “align_optimal()” function. Thanks for pointing out that this essential detail is missing. We would add this content to both the main text and the appendix.

---

> > ### Comment · Reviewer_NnWf · 2025-04-02
> >
> > I appreciate the authors' efforts for resolving and clarifying my concerns. Regarding the question of the difference between efficiency and distinctiveness, I thank you for explaining the difference so clearly, I did not appreciate the effect of codebook size and how that would make both metrics distinct. It may have been my own easy with reading comprehension, but perhaps it could be useful for other readers to spell it out either in the main text or the appendix. However, I leave that decision to the authors' judgment, as it may have been just my own personal problem understanding. Otherwise, I am satisfied with the answer from the authors and the changes they've proposed. I think that the paper is solid, and therefore will keep my current score (4: Accept).

---

> > > ### Author Response · Authors · 2025-04-04
> > >
> > > Dear Reviewer,
> > >
> > > Thanks so much for your support on our project! In the final version, we will include these additional discussions and especially spell out in the main text for the relation between utilization and efficiency according to your great suggestions.
> > >
> > >
> > >
> > > Best,
> > > Authors

---

### Official Review · Reviewer_9RGj · 2025-02-23

**Overall Recommendation:** 3

**Summary:**

The paper presents a new framework, StructTokenBench, for evaluating protein structure tokenization (PST) methods, which break down protein 3D structures into discrete or continuous representations. This framework is critical because existing methods for protein structure tokenization (PST) lacked a unified evaluation system.

Key highlights:

StructTokenBench evaluates PSTs across four axes: effectiveness, sensitivity, distinctiveness, and efficiency. The framework focuses on local, fine-grained protein substructures, unlike typical benchmarks that focus on global structures.

AminoAseed is introduced as an improved method over existing VQ-VAE-based PSTs. It addresses the issue of "codebook collapse," where many codes in the latent space remain unused. The proposed solution introduces codebook reparameterization and Pareto-optimal codebook configurations. These methods help increase the efficiency and utilization of the codebook.

Benchmarking Results: The paper compares AminoAseed with other leading PSTs like ESM3, ProTokens, FoldSeek, and ProteinMPNN, showing a performance improvement on some tasks

Challenges Identified: Current PST methods, especially VQ-VAE-based approaches, struggle with efficient codebook utilization, as large portions of the codebook remain unused, leading to reduced model expressiveness.

AminoAseed Performance: The proposed method performs well in tasks that require a high sensitivity to structural changes and achieves significant improvements in codebook utilization.

**Claims And Evidence:**

Yes.

**Essential References Not Discussed:**

N.A.

**Experimental Designs Or Analyses:**

Yes.

**Methods And Evaluation Criteria:**

Yes.

**Other Comments Or Suggestions:**

N.A.

**Other Strengths And Weaknesses:**

AminoAseed does not perform well on some tasks such as Binding Site Prediction as indicated in Figure 4.

Although the paper demonstrates the benefits of scaling the model (e.g., increasing codebook size or encoder size), it also shows diminishing returns as the model scales. Large-scale models still encounter sub-exponential improvements, meaning that simply increasing compute resources or dataset size does not guarantee a proportional performance boost.

The Pareto-optimal codebook configuration and codebook reparameterization strategies, while innovative, add complexity to the model. The additional computational overhead could be a disadvantage in real-world applications where time and resources are constrained, and simpler methods may be preferred for quick prototyping or high-throughput applications.

Structural tokenization methods, including AminoAseed, tend to lose reliability under high noise levels in the input protein data.

**Questions For Authors:**

It was noted that AminoAseed performed well in structural tasks but not as well in sequence-based tasks like remote homology detection. Do you plan on integrating AminoAseed with sequence models to improve its performance on such tasks? If so, how would you approach this integration?

The study suggests diminishing returns when scaling up the model. In your opinion, what would be the next steps for scaling up AminoAseed more effectively? Do you believe that improvements to the architecture or optimization strategies could yield better performance with additional compute?

**Relation To Broader Scientific Literature:**

Yes.

**Theoretical Claims:**

N.A.

---

> ### Author Rebuttal · Authors · 2025-04-01
>
> Thanks for invaluable suggestions. We respond to your questions below
>
> >R1: AminoAseed doesn’t perform well on binding site prediction in Fig. 4
>
> We'd like to clarify our observation:
> 1. **Fig. 4 shows that when using continuous structural representations**, AminoAseed beats ESM3 on all three binding site prediction tasks, while being comparable to the continuous version of VanillaVQ on two of the tasks.
> 2. **When using discrete structural tokens**, which is the most important application setting for PSTs, AminoAseed prevails for most of the tasks, and gains improvement for averaged task perf. (in Tab. 2)
>
>
> >R2: Paper shows the benefits of scaling the model (e.g., increasing codebook size or encoder size) and the diminishing returns as model scales
>
> We'd like to clarify this rephrased statement from the paper:
> 1. We show in paper the benefits of data-driven Pareto-optimal scaling (i.e., balancing) for the codebook size and codebook dimension. It’s important to notice that we kept the total codebook parameters the same
> 2. We show the diminishing sub-exponential returns in Fig. 7 when increasing encoder sizes, keeping the codebook configuration unchanged
>
> >R3: Pareto-optimal codebook configuration and codebook reparametrization, while innovative, add complexity to the model
>
> We'd like to clarify that **only codebook configuration adds complexity** due to multiple runs of model training, while **reparameterization shares the same complexity** as the ablation model VanillaVQ
>
> In general, for the codebook configuration, **we leveraged the additional resources for exploration and provide insights for others** who aim to develop better PSTs: under a fixed computational resources, balancing codebook sizes and dimensions could be useful. This is a discovery that’s previously unknown in the field
>
> Simpler methods without using more computational resources are indeed favored, and we would like to explore it in our future work
>
> >R4: PST methods, including AminoAseed, tend to lose reliability under high noise levels in the input protein data
>
> We would like to clarify that in Fig. 5, the noise is added to the induced structural representations, not on the raw protein structures. This is achieved by replacing them with the [Mask] representation under a given mask rate.
>
> Essentially, it’s expected to observe large performance degradation at high noise levels (up to 90% tokens masked), since most structural information is removed
>
> >R5: AminoAseed does not do well in sequence-based tasks like remote homology detection
>
> We would like to clarify that the remote homology detection task in StructTokenBench (i.e., “Homo” task) **is not a sequence-based task, but a structure-based task**. Namely, we use backbone structure as input, and do not leverage the residue sequences. As explained in App. A.1.1, our “Homo” task is from the same source TAPE as the popular sequence-based remote homology detection task, which might be the reason for the confusion.
>
> Also, **we appreciate the suggestion and would like to humbly point out this statement does not align with our observation**. As shown in Tab. 2, on “Homo” task, **our method AminoAseed performs the best, achieving 27.31% improvement over ESM3**
>
> >R6: Do we plan on integrating AminoAseed with sequence models to improve its performance on remote homology detection? If so, how would we approach the integration?
>
> 1. **We explored the setting of adding sequence information directly to PST** (i.e., using structural representations + sequence token + positional encoding as input, and passing it to the probing layer). As shown in Tab. 5, adding sequence can mostly improve PSTs’ performance, as expected
> 2. Further explore using sequence models like ESM2 on this specific homology task may shifts from our main focus, i.e., to benchmark structural representations induced from PSTs. **We would like to leave it for future work**
> 3. **For the integration method**, one way is to combine the pretrained protein sequence representations from models like ESM2, with AminoAseed’s structural representations. Fusion attention layers can be added after the two models for better combination. Per-residue and per-protein contrastive learning can be used as training objectives.
>
>
> >R7: What’s the next step to scale up AminoAseed more effectively? Improvement to the architecture or optimization strategies could yield better performance with more compute?
>
> 1. **We recommend data scaling**: gathering more diverse pretraining datasets to improve AminoAseed, since its pretraining data includes only around 10% of the PDB database, with 48k protein single chains.
> 2. **Improving the architecture design** of VQVAE encoder might be useful, because the structure encoder is usually SE-(3) invariant and its capacity can be improved with better design.
> 3. **Improving optimization** would be helpful, which shares the same motivation as AminoAseed to use codebook reparameterization to enable better codebook gradient update.

---

### Official Review · Reviewer_gEVF · 2025-03-13

**Overall Recommendation:** 3

**Summary:**

In order to fully evaluate the performance of existing PST methods, the authors constructed a comprehensive evaluation benchmark called StructTokenBench. AminoAseed, a new improvement scheme, is proposed to address the problem of “codebook collapse” in the traditional VQ-VAE method, whose main innovations include codebook reparameterization and pareto-optimal codebook configuration.

**Claims And Evidence:**

- Claim 1: Adding an MLP can alleviate codebook collapse.


Evidence Concern: It is not fully clear why the additional MLP layer effectively propagates gradients to all codebook vectors. The explanation could be enhanced by further theoretical analysis and ablation experiments.

- Claim 2: IF-based methods outperform VQ-VAE–based methods on many supervised downstream tasks.

Evidence Concern: Although experimental results indicate this trend, it raises questions about the advantages of using a VQ-VAE framework and whether the proposed modifications (AminoAseed) sufficiently close this performance gap.

**Essential References Not Discussed:**

The paper does not discuss related work such as AIDO.St, which appears to be relevant in the context of efficient protein structure tokenization. Including a comparison with AIDO.St would provide a more complete picture of the current state-of-the-art.

**Experimental Designs Or Analyses:**

The experimental setup covers multiple supervised tasks and evaluations (effectiveness, sensitivity, distinctiveness, efficiency).

**Methods And Evaluation Criteria:**

The paper proposes modifications including an MLP for codebook reparameterization and a data-driven Pareto-optimal configuration. These modifications are conceptually interesting; however, more detailed descriptions and comparative evaluations (e.g., against related methods such as AIDO.St) are needed to validate their effectiveness.

**Other Comments Or Suggestions:**

Typo: In Section 2.3, **VQ-VAE** in 'As illustrated in Fig. 1(b), VQ-VAE can be summarized as' should be **Inverse-Folding**.

**Other Strengths And Weaknesses:**

Strengths:
- The paper proposes a comprehensive benchmark covering effectiveness, sensitivity, distinctiveness and efficiency.
- The paper tackles an important problem by addressing codebook collapse in VQ-VAE–based PST.
- The introduction of a data-driven codebook configuration is innovative and supported by extensive experiments.

Weaknesses:

- The explanation of how an MLP helps alleviate codebook collapse is insufficiently detailed.
- Experimental results show that IF-based methods outperform VQ-VAE–based ones on many supervised tasks, raising questions about the benefits of the proposed approach.
- The lack of comparison with related methods such as AIDO.St [1] limits the scope of the evaluation.
- The paper does not provide a detailed description or diagram of the AminoAseed model architecture.
- The absence of complete code release undermines the reproducibility and reliability of the results.

[1] Balancing Locality and Reconstruction in Protein Structure Tokenizer.

**Questions For Authors:**

Please see Other Strengths And Weaknesses.

**Relation To Broader Scientific Literature:**

The paper builds on prior work in protein structure tokenization, VQ-VAE, and inverse folding.

**Theoretical Claims:**

There is an implicit theoretical claim that the MLP layer can mitigate codebook collapse by ensuring all code vectors receive gradient updates. However, the paper lacks a rigorous proof or in-depth discussion on this mechanism. Further theoretical justification would strengthen the contribution.

---

> ### Author Rebuttal · Authors · 2025-04-01
>
> We thank the reviewer for the insightful comments
> >R1: The claim “adding “a MLP” to reparametrize the codebook can alleviate codebook collapse” needs theoretical analysis and ablation experiments
>
> 1. We'd like to clarify that in Sec. 4.2 (L246), we **add a simple linear layer** (instead of a MLP) to reparameterize the codebook $C$ as $Q$=Linear($C$), and keep $C$ fixed during training
>
> 2. **Ablation experiments**: VanillaVQ is the ablation model, which shares the same pipeline and configurations as AminoAseed, except using $C$ instead of Q=Linear($C$). Main results can be found in Tab. 2/3/4/5 and Fig. 3/4/5/6/7. Generally, AminoAseed outperforms VanillaVQ across all four benchmarking perspectives
>
> 3. **Theoretical analysis**: We analyze the gradient flow on $Q$ in AminoAseed and $C$ in VanillaVQ, respectively. We will add this theoretical analysis in the appendix to enhance the method design justification
>
>     **(1) Setup**: Assume $C$ of shape (K, D) is fixed, Linear() denotes a linear transformation with weight $W$ of shape (D, D) without a bias, and the reparameterized codebook $Q$=Linear(C)=$C * W$. We denote $L$ as loss, batch size as 1 for simplicity, and $p$ different codes are selected during tokenization with each code selected for once
>
>      **(2) Gradient on $Q$**: For the gradient $\nabla_Q L$, only the $p$ selected rows of this gradient matrix are non-zero. The gradient updates $W$ via the chain rule: $\nabla_W L = C^T\nabla_Q L$, which is now a dense matrix with all non-zero values. Thus, all entries of $W$ are updated, enabling global adjustments to the entire codebook space through $Q=C * W$
>
>      **(3) Gradient on $C$**: if $C$ is trainable, the gradient matrix $\nabla_C L$ only has the $p$ selected rows as non-zero. Thus, unused codes get no updates, leading to limited changes for the entire codebook space and eventually distribution shift (see Fig. 2)
> >R2: Why using the VQ-VAE framework when IF-based PSTs beat VQ-VAE-based methods on supervised tasks? And if AminoAseed sufficiently closes the perf. gap?
>
> 1. IF-based PSTs excel in supervised tasks but lack sensitivity to conformational changes, limiting their overall capability.
>
> 2. IF-based PSTs only support continuous tokens, while VQ-VAE-based PSTs produce both continuous and discrete tokens. As said in App. C, discrete tokens offer several advantages:
>
>     **(1) Multimodal LLM integration**: enabling seamless fusion with sequence and functional text data, allowing direct use of NLP optimization techniques developed for LLMs.
>
>     **(2) Simplified structure modeling**: eliminating the need to explicitly encode symmetry and physical constraints
>
>     **(3) Reduced overfitting risk**: discrete representations may help mitigate overfitting compared to continuous features [a]
>
> 3. **We acknowledge that there is still much more room for further improvement to close the performance gap** between AminoAseed and IF-based PSTs on supervised tasks. AminoAseed proposes a simple yet effective strategy for improvement over vanilla VQ-VAE, and enjoys the benefits of VQ-VAE methods and shows promising results in “sensitivity” evaluation
>
> [a]DeProt: Protein language modeling with quantized structure and disentangled attention
> >R3: Benchmarking AIDO.st
>
> As suggested, we add AIDO.st and report the results below (some supervised tasks are not finished due to the rebuttal timeline). As shown, AIDO.st is less effective than AminoAseed on supervised downstreams, and also fall short in “sensitivity”. However, AIDO.st achieves very high codebook utilization
>
> |**AIDO.st Effectiveness**|Con|Rep|BindInt|CatInt|BindBio|CatBio|
> |---|---|---|---|---|---|---|
> |Fold|56.64|77.69|44.66|57.30|65.50|73.72|
> |SupFam|73.79|78.08|84.21|81.94|66.70|78.66|
>
> |**Sensitivity**|Metric|**AIDO.st**|
> |---|---|---|
> |apoholo|pearsonR|	43.02|
> ||spearmanR|54.25|
> |foldswitching|pearsonR|61.59|
> ||spearmanR|66.11|
>
> |**AIDO.st Codebook Utilization**|UR%|Perplexity|Entropy|
> |---|---|---|---|
> |CASP14|88.05|0.7729|0.01165|
> |CAMEO|95.12|0.8266|0.01181|
> >R4:  Detailed description or diagram of the AminoAseed model architecture are not provided
>
> 1. We provided the **detailed description** for AminoAseed **in App. E**, including **(1)** overall pipeline of protein frame input, encoding, tokenization, decoding, and structure reconstruction objectives; **(2)** the geometric self-attention layer used in encoder; **(3)** explanation of straight through estimator of gradients in tokenization; **(4)** the standard Gram-Schmidt algorithm to create protein frames as input
>
> 2. We will add a **diagram** for AminoAseed in appendix for next edition
>
> >R5: The absence of complete code release undermines the reproducibility.
>
> We apologize for not mentioning it in the paper. **We will release all the code and processed data in the next months**, because we’re currently in the process of code cleanup and preparation.
> >R6: Typo of in Sec. 2.3
>
> Thanks for spotting the typo and we will modify it

---

### Official Review · Reviewer_LsQQ · 2025-03-14

**Overall Recommendation:** 4

**Summary:**

This paper presents a benchmark for comparing methods of tokenizing proteins. They divide structure tokenization methods into two categories: those which hand-design structure based tokens, and those which learn the tokenization. Of the learned methods, they distinguish between those that produce learned codebooks, and those that use inverse folding (i.e., the “tokenization” is a sequence of amino acids that are supposed to fold into the given backbone). They suggest evaluating methods along four axes: (1) Effectiveness as an input to supervised learning tasks, (2) sensitivity to distinguish similar structures, (3) distinctiveness of codebook vectors, and (4) efficiency, i.e. how uniformly the different codebook elements are used. Evaluation of (1), effectiveness, is the most involved, and entails training a 2-layer MLP on the structure tokens (to which a positional encoding is added) for a variety of different supervised tasks, including prediction of binding sites, conserved sites, and catalytic sites. Applying their benchmark to baselines, they find that some under-utilize the learned codebooks. As a result, they define a new tokenization method, “AminoAseed”, that (1) uses reparametrization during the codebook gradient update to prevent “codebook collapse, and (2) a data-based heuristic for trading off the number of codebook elements and the dimension of each element. AminoAseed does better on the new benchmark than existing approaches. Finally, they conduct ablations and scaling experiments.

**Claims And Evidence:**

For the most part, yes, but with a few minor exceptions below:

The claim that their benchmark focuses on “fine-grained local substructures rather than global structures, as typical in existing benchmarks” (from the abstract). It wasn’t clear to me how this is true — is it due to the nature of the supervised tasks chosen for the “efficiency” section? I recommend this be explained more explicitly in the main body, if the claim is made in the abstract.

There are also claims about the importance of distinctiveness (how different the codebook elements are from one another) and efficiency (how much the different codebook elements appear), which are used to motivate these aspects of the benchmark, but Section 6 (L415) itself shows that efficiency and reconstruction accuracy aren’t correlated. Shouldn’t this call into question the efficiency metric? L179 also makes a claim about the utility of distinctiveness in downstream tasks, without a reference/citation to support it.

**Essential References Not Discussed:**

An essential reference not discussed in detail is “Tokenized and Continuous Embedding Compressions of Protein
Sequence and Structure” by Lu et al 2024. The reference is noted in the appendix, but not the main body. I presume this is because it is an all-atom tokenization, i.e. it essentially includes the residue information (not just the backbone), and indeed, it takes the amino acid sequence as input. However, I think this is a key reference because their evaluations are aimed at comparing different protein tokenizations, and also have a similar narrative of “finding and correcting an existing problem in a PST”.

In addition, the paper “BindGPT: A Scalable Framework for 3D Molecular Design via Language Modeling and Reinforcement Learning” by Zholus et al 2024 is one example of a paper that uses a simple atom-wise coordinate tokenization, essentially just tokenizing the coordinates as numbers. (Similarly, Geo2Seq in “Geometry Informed Tokenization of Molecules for Language Model Generation” by Li et al 2024 tokenizes 3D coordinates, but in spherical coordinates.) These kinds of simple methods — which are inefficient but very accurate as compressors — should be benchmarked as well.

**Experimental Designs Or Analyses:**

Not in very great detail. However, I do question, in the supervised tasks, why an MLP was used with positional encodings — first, if the intended downstream use case for PSTs are generally LLMs, would it not make more sense to train a transformer? Second, if one is committed to using an MLP, what’s the justification for a positional encoding? An MLP is not permutation invariant.

**Methods And Evaluation Criteria:**

Yes

**Other Comments Or Suggestions:**

Minor writing suggestion: I would suggest more explicitly motivating how PSTs are used downstream, so that each aspect of the benchmark can be traced back to a concrete goal for PSTs. This is currently not clear to me, specifically efficiency and distinctiveness.

Minor writing suggestion: For clarity, I would recommend being more explicit (even in the introduction, or perhaps a footnote/appendix) about when the tokenizations capture backbone only vs residues also. This can be confusing to newcomers, especially distinguishing from representations of structure in the literature that are referred to as “all atom” or “joint sequence and structure”. (For example, what does “amino acid tokens” near L052 refer to?)

Minor writing suggestion: This is subjective, but in my opinion “effectiveness” and “efficiency” are not the most intuitive terms for the attributes the benchmark is measuring. (I often had to check back to figure out which term I wanted to use when writing this review.) Perhaps something like “codebook utilization” instead of “efficiency” would be preferable, or “downstream effectiveness” instead of just “effectiveness”.

L1029: typo, “linear layer, ,”

**Other Strengths And Weaknesses:**

Strengths:

The problem addressed is an important one and addresses a gap in the literature. Having a standardized benchmark for comparing protein tokenization methods (that is relatively independent of the downstream autoregressive model, and in particular does not require training many large models from scratch with different tokenizations to figure out which one is best), is quite useful. The benchmark is thoughtful, covering a wide range of tasks and facets, which have seemingly been implemented with very careful attention to detail. The empirical analyses (ablations and scaling studies) provide valuable insights for future work.

Weaknesses:

First, the paper leaves out several PSTs from its framing and experiments. As mentioned in the “related work” section of the review, there are many structure tokenizations that consist of tokenizing the 3D coordinates of the backbone atoms. Generally speaking, the introduction mentions “heuristic” tokenizations, but as far as I can tell, doesn’t compare to any of them. Also, a tokenization benchmark would be useful for structure tokenizations that include sequence as well (not just backbone), but methods like CHEAP are not included.

Moreover, the “distinctiveness” and “efficiency” desiderata articulated in the benchmark itself — feel a bit ad-hoc to me. This is supported by the paper’s own findings in Section 6, which counterintuitively show that reconstruction accuracy and efficiency are not correlated.

In fact, the “efficiency” metric is seemingly quite distribution-dependent. A PST might induce a uniform distribution over tokens on one set of proteins, but a non-uniform distribution on a very different data distribution. This would not be a failing of the PST, but potentially an ability of it to accurately reconstruct diverse proteins. Indeed, this generalizability is one advantage of the coordinate-wise tokenizations (e.g. used in BindGPT and others), but they may fail at the “efficiency” metric.

A more standard metric from the channel compression formalism of information theory is simply to assess how effective the methods are as compressors, i.e. for a fixed reconstruction accuracy (based on solving the “reconstruct structure from PST” learning task), how many bits are used to represent the structure (on average, on a given dataset)?

Finally, it seems like the benchmark assumes that the proposed probing method is a good proxy for how a PST will perform in downstream tasks, but this is not supported. It would be good to have evidence for this e.g. on a single task, ablating architecture choice and other factors.

Overall, I think this paper still makes very valuable contributions, and recommend acceptance. But if the weaknesses above were addressed, I would probably increase my rating further.

**Questions For Authors:**

1. For the “efficiency” part of the metric, why is the positional encoding necessary when the probing network is an MLP? Positional encodings are traditionally used for transformers, which are permutation invariant by default. On a related note, why use an MLP over a transformer? Wouldn’t we expect a transformer to correspond more faithfully to downstream performance, if most of the PSTs are going to be passed into a transformer-based architecture in practice?

2. Did you consider the computational efficiency of computing the structure-based tokens at all? This could be a future addition to the benchmark, e.g. compute-controlled efficiency. Alternatively, perhaps one cares about compute time in the downstream application (e.g. for a transformer, the context length, which doesn’t intrinsically have to be the number of amino acids).

3. Why do “distinctiveness” and “efficiency” inherently matter? As mentioned above, these concepts might be at odds with faithfully representing structure under distribution shift. If a tokenization is useful for downstream tasks and enables small context window (i.e. the number of tokens is small), why would we care about these specific properties of the codebook elements? Instead, why not just frame as “effectiveness as a compressor” (i.e. expected reconstruction accuracy vs size of compressed representation in bits, or context length)?

4. What is the intention of the noise robustness analysis? It’s not clear to me if we should expect a good tokenization to be robust to generic noise or not. It seems like certain perturbations *should* meaningfully change the PST, whereas others (e.g. if the result is non-physical in some way) should not.

**Relation To Broader Scientific Literature:**

This work is relevant to essentially all prior works that use protein tokenization as a way of dealing with 3D structures in a LLM setting. To my knowledge, there has not been as thorough of a systematic comparison of these methods before, and having a standardized benchmark is quite valuable to the community. AminoAseed builds on ESM3 and other prior VQ-VAE approaches.

**Theoretical Claims:**

n/a

---

> ### Author Rebuttal · Authors · 2025-04-01
>
> Thanks for your valuable comments! We address concerns below
> >R1: Explain “StructTokenBench focuses on local(per-residue) over global(per-protein) structures”
>
> PSTs tokenize per-residue substructures, matching StructTokenBench: **supervised tasks and “Sensitivity” are at residue level**. Current benchmarks(Sec. 6.1) cover per-protein tasks, justifying our per-residue focus. We’ll add in main text
> >R2: No citation for “Distinctiveness” motivation(L179)
>
> **We’ll cite ESM3 Fig. S5**: ambiguous structural token mapping harms. This metric assess the token ambiguity via codebook similarity.
> >R3: Efficiency(Codebook Utilization) metric seem conflicting with Sec 5.7(L415): no correlation with reconstruction quality. Is it needed?
>
> We rename “Efficiency” to “Codebook Utilization”(% of used codes). Underutilized codebooks waste resources and harm downstream perf.(Sec. 3.4), a well known issue in CV/NLP[a]. High utilization bring gains, like LLaMa3’s 128K-token vocabulary for efficient encoding[a]. **Sec. 5.7 questions if reconstruction quality alone reliably indicates PST quality**, given no correlation with **both supervised task perf. and utilization**(see Fig. 6,12) We welcome further discussion
> [a]LLama technical report
> >R4: Codebook utilization is distribution-dependent. PSTs' non-uniform token use may reflect accurate reconstruction(like BindGPT)
>
> We agree on dependence. Current results hold across datasets. We’ll add more diverse datasets. We remain inconclusive on exact utilization-reconstruction relations. **Shown in Sec. 5.7**, we question if reconstruction quality alone reliably indicates PST quality, given no correlation with **both supervised task perf. and utilization**. We welcome further discussion. Thanks but we can’t test it as it lacks residue-level structural reprs.(see **R8**)
> >R5: Why use probing for supervised tasks? Why MLP not transformer?
>
> Probing(fixed encoder+simple probing layer) is standard practice in CV[b]. Being simple ensures perf. reflects reprs. quality, not probing layer capacity. For BindInt, empirical perf. below show: linear layer lags 2-layer MLP, which matches 1-/2-layer transformer perf. Limited dataset sizes(Tab. 6,8) may explain transformers' marginal gains
> |Method|Split|Linear|2L MLP|1L Transformer|2L Transformer|
> |---|---|---|---|---|---|
> |**ESM3**|Fold|43.36|44.30|46.78|46.08|
> ||SupFam|84.50|90.77|90.98|90.96|
> [b]Masked Autoencoders Are Scalable Vision Learners
> >R6: Why use positional encodings(PEs), as MLP isn’t permutation-invariant?
>
> Our MLP is shared across structural tokens, so **is permutation-equivariant**. So it needs PEs for residue order. Empirically, removing PEs largely hurts perf.(see table below vs. Tab. 2)
> |ESM3 no PEs|BindInt|CatInt|Con|Rep|
> |---|---|---|---|---|
> |Fold|48.85|53.14|51.05|51.36|
> |SupFam|74.16|69.42|62.03|62.15|
> >R7: Test Cheap
>
> It’s initially excluded as it models **sequence and all-atom structure**(37 atoms/residue), while our benchmark only considers backbone(4 atoms/residue, Sec. 2.1). Cheap’s perf.(vs. Tab. 2,3) is in **https://anonymous.4open.science/r/cheap_perf-00EB** limited by space: (1)it beats IF-based PSTs on supervised tasks (some results pending) (2)It largely lags AminoAseed in “sensitivity”. (3)No other evaluation as its has no codebook
>
> >R8: Test BindGPT, Geo2Seq
>
> Thanks for referring to atom-wise coordinate tokenization methods. We’ll discuss in main text and appendix. BinGPT is untestable as: (1)It encodes atom types and xyz coordinates as strings. Unlike IF-/VQ-VAE PSTs, it doesn’t output residue-level structural reprs; (2)Being decoder-only, BindGPT also can’t produce residue-level reprs. Geo2Seq is not open-source
> >R9:Heuristic PSTs not tested
>
> We’ll add them later due to deadline
> >R10:For fixed reconstruction quality, measure bits used
>
> We use scatter plot: X-axis: reconstruction quality(RMSD or LDDT),Y-axis: Compression ratio(% byte reduction vs. XYZ coordinates).**Plots in https://anonymous.4open.science/r/rebuttal_fig-3050**. AminoAseed beats VanillaVQ in reconstruction but trails ESM3, while achieving better compression(smaller byte reduction %) than both
> >R11: Explicitly motivate PSTs
>
> We’ll add**R3/R4** discussions on distinctiveness and codebook utilization in main text
> >R12:State PSTs' modeling targets
>
> We’ll add **(1)Tab. 2 footnote**: Cheap(sequence + all-atom structure) vs. others( backbone structure) and **(2)appendix table**: PST comparison (backbone/all-atom, sequence usage) including non-open-sourced
> >R13: Re-term
>
> We change “efficiency” to “codebook utilization” and “effectiveness” to “downstream effectiveness” for clarity
> >R14:Computational efficiency as future work
>
> Will leave for future
> >R15: Intention of noise analysis
>
> Noise is added by randomly masking structural tokens. We reveal: (1) Structural reprs. are less robust than sequence reprs., **so integrating sequence-structure may help** (2) Practical guidance **for masked language modeling on structural tokens**
> >R16:Typo in L1029
>
> Will modify

---

### Decision · Program_Chairs · 2025-05-01

**Decision:**

Accept (poster)

**Comment:**

This paper aims to benchmark different protein tokenization methods. To this end, the authors introduce **StructTokenBench**, a comprehensive framework for evaluating existing tokenization approaches. Additionally, they propose **AminoAseed**, a simple yet effective strategy that improves codebook gradient updates while optimally balancing codebook size and dimensionality, resulting in better tokenizer utilization and higher quality.

The paper is well-executed, and all reviewers voted for acceptance following the rebuttal.